# FADS1/2 control lipid metabolism and ferroptosis susceptibility in triple-negative breast cancer

Nicla Lorito [1,7], Angela Subbiani [1,7], Alfredo Smiriglia[1], Marina Bacci [1], Francesca Bonechi[1], Laura Tronci[2], Elisabetta Romano [1], Alessia Corrado[3], Dario Livio Longo [3], Marta Iozzo [1], Luigi Ippolito [1], Giuseppina Comito[1], Elisa Giannoni [1], Icro Meattini[1,4], Alexandra Avgustinova[5,6], Paola Chiarugi [1], Angela Bachi [2] & Andrea Morandi [1✉]

## Abstract

**Triple-negative breast cancer (TNBC) has limited therapeutic options, is highly metastatic and characterized by early recurrence. Lipid metabolism is generally deregulated in TNBC and might reveal vulnerabilities to be targeted or used as biomarkers with clinical value. Ferroptosis is a type of cell death caused by iron-dependent lipid peroxidation which is facilitated by the presence of polyunsaturated fatty acids (PUFA). Here we identify fatty acid desaturases 1 and 2 (FADS1/2), which are responsible for PUFA biosynthesis, to be highly expressed in a subset of TNBC with a poorer prognosis. Lipidomic analysis, coupled with functional metabolic assays, showed that FADS1/2 high-expressing TNBC are susceptible to ferroptosis-inducing agents and that targeting FADS1/2 by both genetic interference and pharmacological approach renders those tumors ferroptosis-resistant while unbalancing PUFA/MUFA ratio by the supplementation of exogenous PUFA sensitizes resistant tumors to ferroptosis induction. Last, inhibiting lipid droplet (LD) formation and turnover suppresses the buffering capacity of LD and potentiates iron-dependent cell death. These findings have been validated in vitro and in vivo in mouse- and human-derived clinically relevant models and in a retrospective cohort of TNBC patients.**

**Keywords** Ferroptosis; Lipid Metabolism; Polyunsaturated Fatty Acids; Desaturases; Lipid Droplets
**Subject Categories** Autophagy & Cell Death; Cancer; Metabolism

## Introduction

Altered lipid metabolism is generally considered a common feature of breast cancers (Bacci et al, 2021). However, important differences have emerged in terms of lipid synthesis, upload, storage, and utilization between breast cancers with different biology, invasiveness, response to therapy, and prognosis (Koundouros and Poulogiannis, 2020). Intracellular lipids have a role as energetic substrates, structural components, or signaling molecules. Major components of lipids are fatty acids (FA) and their role within the cells is not only dependent on the total amount available but also on their length and complexity (i.e., the presence of carbon-carbon double bonds). Indeed, the balance between saturated (SFA), mono (MUFA), and polyunsaturated FA (PUFA) is essential for cellular homeostasis and biological functions (Dyall et al, 2022). The introduction of the first double bond is mediated by stearoyl-coenzyme A (stearoyl-CoA) desaturase 1 (SCD1), the key rate-limiting enzyme responsible for MUFA formation. Subsequent double bonds are inserted by fatty acid desaturases 1 and 2 (FADS1/2) (Vriens et al, 2019; Xuan et al, 2022), which are thus key mediators of PUFA biosynthesis. Aberrant lipid desaturation is involved in tumor initiation and progression and can be effectively targeted to impair tumor growth, metastatic dissemination, and relapse in preclinical models (Li et al, 2017; Peck et al, 2016; Ran et al, 2018). Particularly, PUFA have a major role in maintaining the architecture of the plasma membrane (Harayama and Shimizu, 2020). When in excess, PUFA are rapidly incorporated into the plasma membrane by long-chain acyl-CoA synthetase 4 (ACSL4) (Doll et al, 2017) and become vulnerable to lipid peroxidation. Peroxidized lipids alter the membrane bilayer, making it susceptible to rupture and causing ferroptosis, a unique form of iron-dependent cell death (Dixon et al, 2012). Consequently, the FA composition of the membranes, particularly a higher presence of PUFA over that of MUFA, enhances ferroptosis sensitivity (Lei et al, 2022). Lipid droplets (LD) can serve as a lipid reservoir for maintaining the intracellular balance of PUFA and MUFA, hence playing an essential role in ferroptosis protection (Dierge et al, 2021). Recently, many cell-autonomous pathways that prevent or revert lipid peroxidation have been shown to counteract ferroptosis induction. Among the mechanisms described, the antioxidant activity of the membrane-associated glutathione peroxidase 4 (GPX4) and the increased availability of the scavenger

[1]Department of Experimental and Clinical Biomedical Sciences, University of Florence, Viale Morgagni 50, 50134 Florence, Italy. [2]IFOM ETS - The AIRC Institute of Molecular Oncology, Via Adamello 16, 20139 Milan, Italy. [3]Institute of Biostructures and Bioimaging (IBB), National Research Council of Italy (CNR), Via Nizza 52, 10126 Torino, Italy. [4]Radiation Oncology Unit, Oncology Department, Azienda Ospedaliero Universitaria Careggi, Largo Brambilla 3, 50134 Florence, Italy. [5]Institut de Recerca Sant Joan de Déu, Carrer Santa Rosa 39-57, 08950 Esplugues de Llobregat, Spain. [6]Institute for Research in Biomedicine (IRB Barcelona), The Barcelona Institute of Science and Technology, Baldiri Reixac 10, 08028 Barcelona, Spain. [7]These authors contributed equally: Nicla Lorito, Angela Subbiani. ✉E-mail: andrea.morandi@unifi.it

molecule glutathione (GSH), sustained by the SLC7A11-mediated import of cystine (Koppula et al, 2021), have a prominent role in regulating ferroptosis sensitivity.

Here we describe a cell-autonomous mechanism that exposes aggressive triple-negative breast cancer (TNBC) cells to ferroptosis susceptibility. We found that aggressive TNBC cells are characterized by higher expression of FADS1/2, leading to enhanced PUFA availability and thus becoming more sensitive to ferroptosis-promoting insults, a finding validated both in vitro and in vivo. Indeed, targeting FADS1/2 renders the aggressive TNBC cells resistant to ferroptosis as the addition of exogenous MUFA to the aggressive models exerts an anti-ferroptosis effect and prevents the detrimental effects of lipid peroxidation, whereas increasing the availability of PUFA resensitizes the resilient models to ferroptosis execution. Finally, we show that combining ferroptosis induction with the inhibition of LD biogenesis could be a promising approach for treating TNBC.

# Results

## Altered lipid metabolism is a feature of aggressive TNBC cells

We used isogenic 67NR, 4T07, and 4T1 TNBC cells, which were previously derived from the same primary tumor and characterized by diverse tumorigenic and metastatic capacities (Avgustinova et al, 2016). Specifically, when injected orthotopically, 67NR cells do not metastasize, 4T07 tumors seed micrometastases only in the lung, whereas 4T1 are overtly metastatic TNBC cells (Avgustinova et al, 2016). It has been previously reported that 4T1 cells are characterized by high metabolic flexibility, a trait that may correlate with their increasing aggressive ability (Simões et al, 2015). To identify whether the observed metabolic features were stochastic or rather supported by a general transcriptomic and metabolomic reprogramming, whole genome expression analysis was performed on tumor cells FACS-isolated from 4T1 and 67NR tumors. Subsequent Gene Set Enrichment Analysis (GSEA) confirmed that the HALLMARK_GLYCOLYSIS gene set (MSigDB M5937) was positively correlated (Normalized Enrichment Score, NES = 2.9; False Discovery Rate, FDR = 0.001, Fig. 1B) with the 4T1 transcriptome and surprisingly revealed a positive correlation with several gene sets related to fatty acid (FA) biosynthesis and lipid metabolism (Fig. 1B), indicating that enhanced glycolysis could also contribute to increased citrate production in the TCA cycle, thus promoting de novo lipid production and utilization.

Therefore, to functionally assess the cells' ability to foster lipid biogenesis, we pulsed 67NR, 4T07, and 4T1 cells with $^{14}$C-Glucose and other potential citrate and/or acetyl-CoA precursors (i.e., $^{14}$C-Lactate, $^{14}$C-Acetate, and $^{14}$C-Glutamine) before recovering the radiolabeled lipid fraction. 4T07 and 4T1 cell lines showed a significant increase in lipid biogenesis, irrespective of the carbon donor used, when compared to 67NR cells (Fig. EV1A). However, within the metastatic cell lines, 4T1 cells exhibited a superior capacity of fueling lipogenesis through lactate and acetate, whereas 4T07 cells predominantly utilized glutamine. Accordingly, qRT-PCR and western blot (WB) analyses revealed increased expression levels of the key lipogenic enzymes ATP Citrate Lyase (ACLY) in 4T1 cells and Fatty Acid Synthase (FASN) in both 4T07 and 4T1 cells when compared to 67NR cells (Figs. 1C and EV1B). To exclude that the observed metabolic phenotype was cell

line specific, we validated the findings in two isogeneic spontaneously metastatic sublines, D2A1-m1 and D2A1-m2, generated from the poorly metastasizing murine-derived D2A1 cell line by serial in vivo passaging (Jungwirth et al, 2018). D2A1 metastatic cells showed increased ACLY and FASN expression both at mRNA and protein levels when compared to parental cells (Figs. 1C and EV1C). Importantly, we observed a comparable pattern (Fig. 1C) across a panel of six human TNBC cell lines that have been reported to exhibit a diverse tropism for secondary sites (Fig. 1A). Specifically, BT20 and HCC1937 cells can disseminate to distant sites without substantial proliferation and expansion, while the metastatic potential is evident in MDA-MB-231, MDA-MB-468, and SUM159 cells. Among these, MDA-MB-231 cell line represents the prototype of highly metastatic and penetrant TNBC cells, as reported in previous studies (Jin et al, 2020). This confirmation in human models reinforces the observations made in murine models. Since it has been reported that the luminal androgen receptor (LAR) TNBC subtype is characterized by enriched FA metabolism and lipogenic phenotype (Yang et al, 2023), we also validated our initial findings in the human MDA-MB-453 LAR metastatic cell line (Fig. 1A,C).

This enhanced ability to sustain lipogenesis was paralleled by an increased lipid accumulation in form of lipid droplets (LD) content in the more aggressive 4T07, 4T1, D2A1-m2, MDA-MB-468, MDA-MB-231, and SUM159 cells as visualized by confocal microscopy (Fig. 1D) and/or flow cytometry (Fig. EV1D) analyses using the fluorescent neutral lipid dye BODIPY$^{493/503}$. LD increased accumulation was also supported by increased levels of protein and mRNA expression of Perilipin 2 (PLIN2), a key protein responsible for LD stability and formation (Figs. 1E and EV1E).

Although maintaining a high rate of de novo FA synthesis concomitantly to lipid breakdown (i.e., fatty acid β-oxidation, FAO) could be biochemically and energetically unproductive, it has been reported that this apparently futile metabolic loop can occur in breast cancer cells that survive tumor regression (Havas et al, 2017). To investigate the nutrient contribution to the oxidative metabolism of TNBC cells, we real-time measured the oxygen consumption rate (OCR) in 67NR, 4T07, and 4T1 cells using Seahorse analysis (Fig. EV1F,G). Mitochondrial Stress Test was performed either in the presence of a mitochondrial pyruvate carrier inhibitor (UK5099), of a glutaminase inhibitor (BPTES), or by blocking lipid entry into the mitochondria using the carnitine palmitoyltransferase 1A (CPT1A) inhibitor etomoxir. This approach allows for dissecting the oxidative contribution of glucose-derived pyruvate, glutamine, and FA, respectively. Although UK5099 had the maximal OCR inhibition in the three cell lines (i.e., glucose dependency, Fig. EV1F), etomoxir significantly and selectively impaired basal and maximal respiration in both 4T07 and 4T1 cells (Fig. EV1G), a dependence that was validated in 4T1 cells using the Oroboros oxygraph-2K high-resolution respirometer (Fig. EV1H), indicating that the catabolic contribution of lipids is higher in the more aggressive models.

To identify potential metabolic vulnerabilities associated with the lipid metabolic reprogramming displayed by the more aggressive TNBC cells, we comparatively assessed the changes in cell viability induced in 67NR and 4T1 cells exposed to agents that target lipid-related metabolic pathways. Unexpectedly, targeting ACLY (SB-204990), Acetyl-CoA Carboxylase 1 (ACC1, TOFA), FASN (TVB3166), FAO (etomoxir), LD biogenesis (Diacylglycerol Acyltransferase 1 and 2, DGAT1 and DGAT2, inhibitors) or lipid

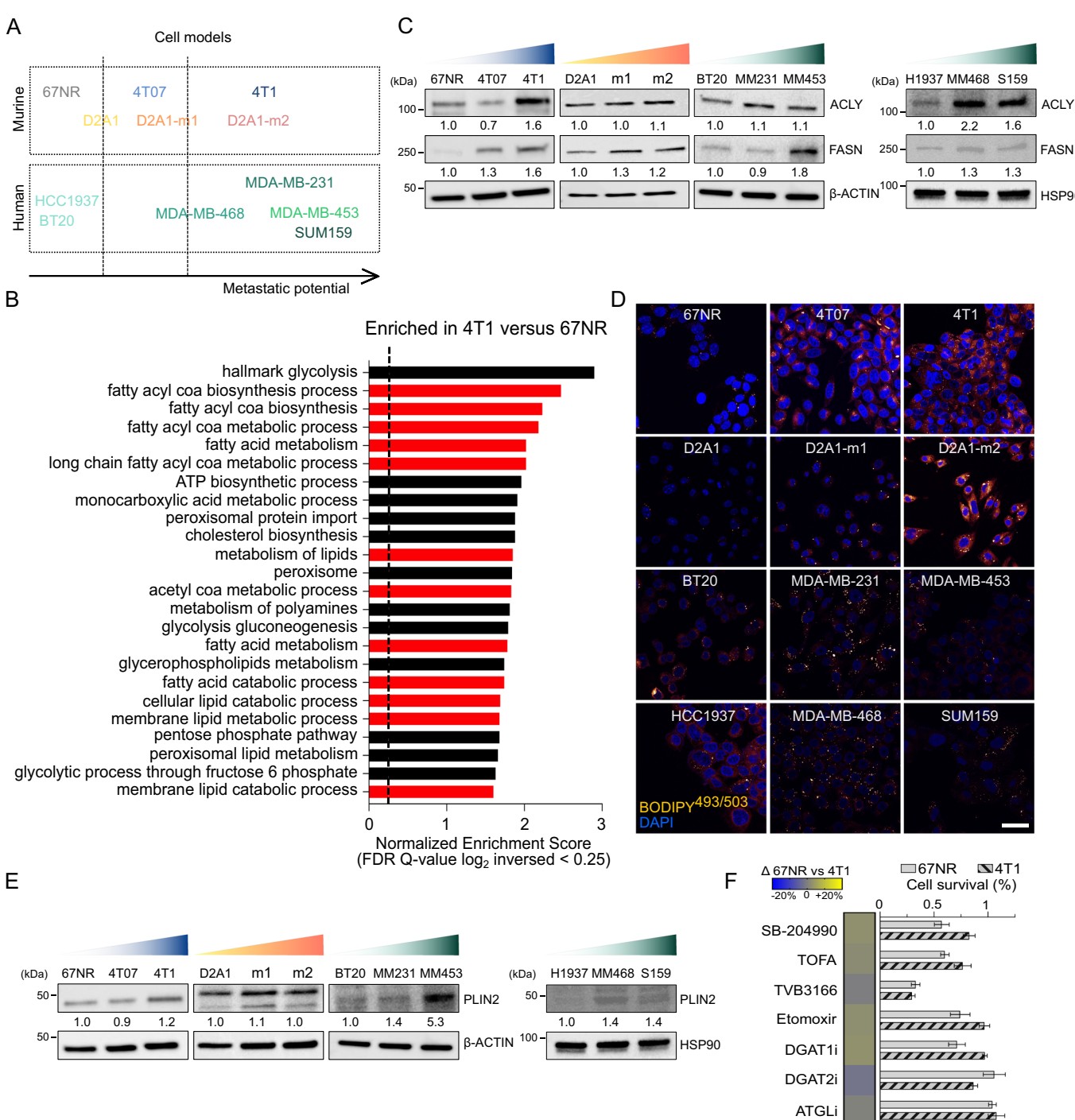

mobilization from LD (Adipose Triglyceride Lipase, ATGL, inhibitor, ATGListatin) showed no significant difference between 67NR and 4T1 cells (Fig. 1F).

## Aberrant lipid metabolism sensitizes aggressive TNBC cells to ferroptosis

Based on this evidence, we hypothesized that the lipid metabolism of the more aggressive cells does not impact the total amount of intracellular lipids but rather their composition and complexity, a proxy of desaturase activity. Indeed, the production of MUFA is catalyzed by SCD, whereas FADS are responsible for subsequent double bond insertion and PUFA formation (Peck and Schulze, 2016). Western blot analysis revealed increased expression levels of either SCD1 or SCD2 together with FADS1 and FADS2 in the 4T07 and 4T1 cells (Fig. 2A). A similar trend was observed for FADS1 and FADS2 protein or mRNA level in D2A1 and human metastatic models, although the expression levels of SCD1 and SCD2 in human cell lines exhibited a heterogenous pattern (Figs. 2A and EV2A). The clinical relevance of these findings was validated

**Figure 1. Altered lipid metabolism is a feature of aggressive TNBC cells.**

(A) Schematic representation of the metastatic potential of 4T1, D2A1, and human TNBC cellular models. 67NR, 4T07, and 4T1 cells (referred to as the 4T1 series) originated from a spontaneous mammary adenocarcinoma growing in a BALB/cFC3H mouse but have undergone different selection processes and exhibited distinctive aggressiveness and metastatic potential. Highly metastatic D2A1-m1 and D2A1-m2 cells have been selected from the weakly metastasizing murine-derived D2A1 cells by serial in vivo passaging. Human TNBC cell lines with different metastatic potentials have been employed. These cellular models are here represented subdivided into three main clusters: weakly, intermediate, and highly metastatic. (B) Gene Set Enrichment Analysis (GSEA) results of the top gene sets showed a positive association with the 4T1 versus 67NR gene expression profile (GSE236033). FDR, false discovery rate. (C) Total protein lysates from weakly to highly metastatic murine and human breast cancer cells were subjected to WB analysis with the antibodies indicated. (D) Murine and human TNBC cells were subjected to confocal analysis. Representative pictures of BODIPY$^{493/503}$ stained cells are shown (orange/yellow: LD; blue: DAPI, nuclei; scale bar, 10 μm). (E) Murine and human breast cancer cells were analyzed by WB analysis using the antibodies described in the figure. The loading controls of the human cell lines (β-ACTIN) in Figs. 1C, E and 2A and (HSP90) in Figs. 1C and 2A are the same and derived from the same experiment (i.e., cell lysates) and blots were processed in parallel. (F) 67NR and 4T1 cells were treated with a series of drugs targeting lipogenic enzymes and subjected to cell viability assays. The heatmap showed that no drug exerts a differential effect between the two cell lines ($n = 3$ biological replicates in either single, duplicate, or technical triplicate). Data information: In F data are represented as mean ± SEM, two-way ANOVA, Bonferroni corrected. Source data are available online for this figure.

retrospectively in a cohort of TNBC. TNBC-bearing patients were dichotomized based on higher and lower mRNA expression of both *FADS1* and *FADS2* (i.e., FADS1/2$^{high}$ and FADS1/2$^{low}$ expressing patients). High expression of both desaturases correlates with reduced relapse-free survival (RFS, hazard ratio [HR] = 1.4, log-rank $P = 0.04$, $n = 534$, Fig. 2B) and overall survival (OS, HR = 2.89, log-rank $P = 0.026$, $n = 126$, Fig. 2B) in TNBC patients. In line with the characterization of the in vitro metastatic TNBC models, we also found that high expression of *ACSL4* (HR = 1.59, log-rank $P = 0.096$, $n = 181$; Fig. EV2B), involved in the conversion of FA to their active form acyl-CoA, and *PLIN2* (HR = 1.73, log-rank $P = 0.00032$, $n = 534$; HR = 1.85, log-rank $P = 0.00076$, $n = 404$, Fig. EV2B), a proxy of LD content, correlates with impaired RFS and distant metastasis free survival (DMFS) in TNBC patients.

In addition, by accessing the transcriptomic profile of the FUSCCTNBC cohort (Jiang et al, 2019) which comprises 465 TNBC patients (with RNA-seq data available for 360 of them), we investigated the contribution of *FADS1* and *FADS2* in the different TNBC subtypes. Notably, *FADS1* is expressed at higher levels in LAR and basal-like immunosuppressed (BLIS) subtypes when compared to mesenchymal (MES) and immunomodulatory (IM), whereas *FADS2* is present at the highest levels in LAR subtype when compared to all the other TNBC subtypes, thus confirming that the LAR subtype is characterized by overall higher *FADS* expression levels (Fig. 2C). Importantly, there was a positive correlation between the expression levels of *FADS1* and *FADS2* both in the whole cohort of TNBC and particularly in the LAR subtype (Fig. 2D). Together, these findings demonstrate that FADS1/2 expression has a prognostic value in TNBC patients.

However, unexpectedly, the administration of either SC-26196, a selective FADS2 inhibitor, CP-24879, a FADS1/FADS2 dual inhibitor, or CAY10566, a SCD1 inhibitor, at concentrations able to inhibit the desaturases activity and not be per se antioxidant (Lee et al, 2020), did not affect cell survival of 4T07 and 4T1 cells (Fig. EV2C).

Therefore, we postulated that impacting the availability of PUFA could be a metabolic vulnerability when aggressive cells are exposed to a secondary hit. Indeed, it is established that peroxidation of PUFA drives ferroptosis (Yang et al, 2016) and that MUFA have a ferroptosis-protective effect by suppressing lipid reactive oxygen species (ROS) accumulation at the plasma membrane (Magtanong et al, 2019). Accordingly, it is reported that FADS expression and activity determine ferroptosis sensitivity in gastric and ovarian cancer (Lee et al, 2020; Xuan et al, 2022), whereas those of SCD

sustain ferroptosis resistance in breast cancer (Luis et al, 2021), reinforcing the idea that PUFA intracellular availability is necessary for ferroptosis execution. In mammalian membranes, PUFA within the phospholipids are mainly phosphatidylcholine (PC) and phosphatidylethanolamine (PE) (van der Veen et al, 2017). Total lipids extracted from 4T1 and 67NR cells were subjected to mass spectrometry analysis [nano-liquid chromatography-electrospray ionization-tandem mass spectrometry (nLC-MS/MS)] (Cattaneo et al, 2021). Lipids composition analysis (Dataset EV1) identified the PUFA/MUFA ratio of PC derivatives lysophosphatidylcholine (LPC) to be significantly higher in the 4T1 cells than in 67NR, and a similar trend was also observed for lysophosphatidylethanolamine (LPE) (Fig. 2E), a trait that suggests the 4T1 cells as potentially susceptible to ferroptosis-inducing insults. Notably, 4T07 and 4T1 cells showed higher cytosolic and/or mitochondrial ROS basal content compared to 67NR cells, as revealed by different redox-sensitive probes (Fig. EV2D), a trait that could be related to their enhanced maximal oxygen consumption rate (i.e., mitochondrial oxidative phosphorylation, Fig. EV1F–H). ACSL4 is crucial for PUFA-mediated ferroptosis execution, and its protein and/or mRNA expression levels are higher in the more aggressive murine and human cells and in TNBC patients with poorer prognosis (Figs. 2F and EV2B,E). To note, ACSL5 has recently emerged as a novel ferroptosis-related gene (Zhang et al, 2023) and could represent a possible substitute in cells that do not express ACSL4 (i.e., MDA-MB-453 and MDA-MB-468, Fig. 2F). Therefore, this molecular, metabolic, and redox cellular asset could render the more aggressive TNBC cells highly susceptible to ferroptosis induction. We, therefore, subjected the 4T1 series (Fig. 2G) and the D2A1 series (Fig. EV2F) to increasing doses of the ferroptosis inducer RSL3 (RAS-selective lethal), which inhibits the Glutathione Peroxidase 4, GPX4 (Yang et al, 2014). To exclude a restricted RSL3-mediated effect, the 4T1 and D2A1 series were also exposed to increasing doses of erastin (Fig. EV2G), a ferroptosis inducer that inhibits the cystine/glutamate antiporter (system XC$^-$) known to provide cystine (i.e., the oxidized form of cysteine) required for glutathione synthesis (Dixon et al, 2012). Both RSL3 and erastin reduced cell survival in the more aggressive cells in a dose-dependent manner, whereas the less aggressive 67NR and D2A1 were much more resistant to ferroptosis induction (Figs. 2G and EV2F,G).

Since ROS production and subsequent lipid peroxidation are both hallmarks of ferroptosis, we evaluated upon RSL3 and/or erastin treatment (i) the ratio of reduced and oxidized glutathione

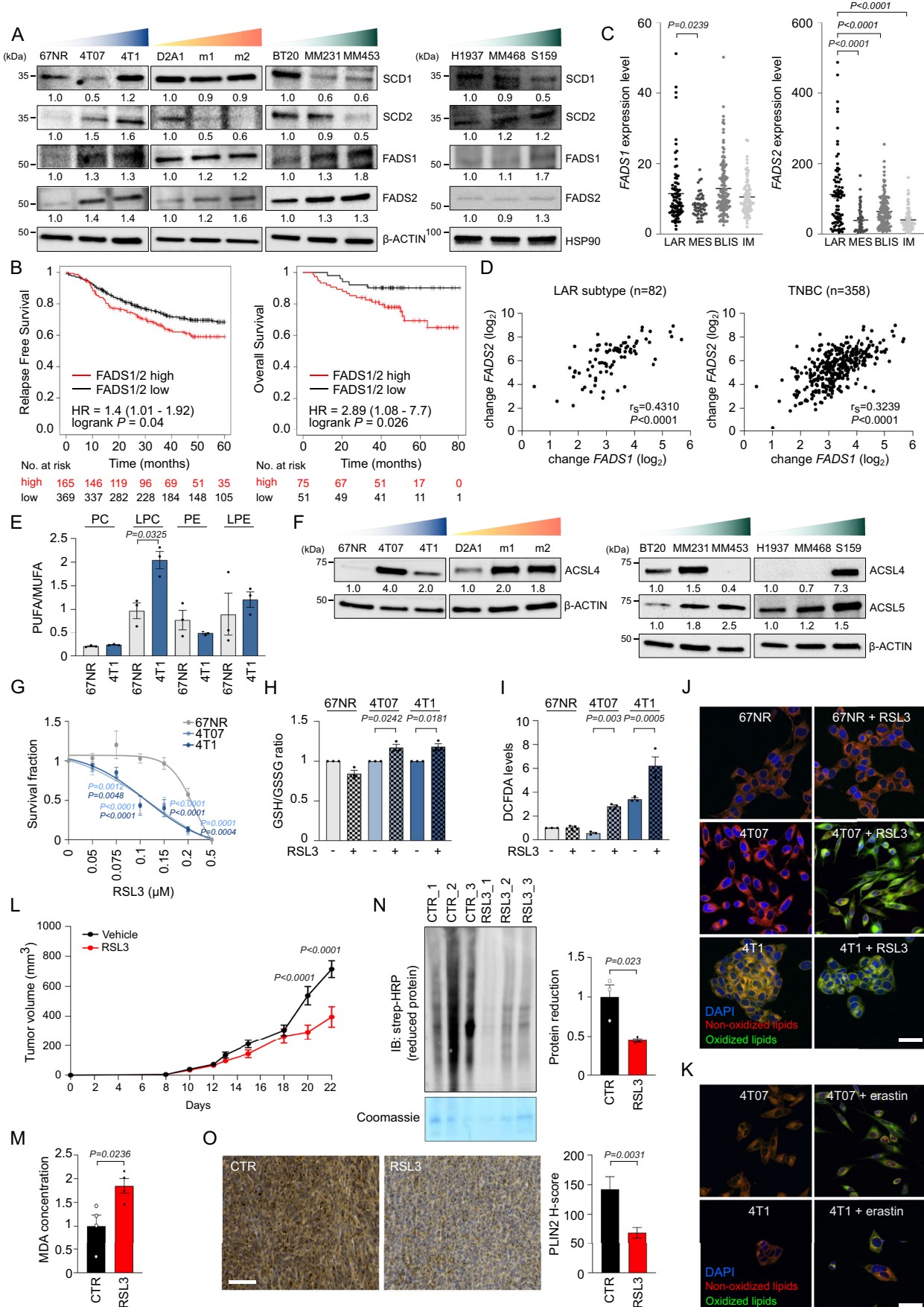

**Figure 2. FADS1 and FADS2 identify aggressive TNBC sensitive to ferroptosis induction.**

(A) Murine and human breast cancer cells, ordered from left to right according to their metastatic potential, were analyzed by WB analysis using the antibodies described in the figure. The loading controls of the human cell lines (β-ACTIN and HSP90) in Figs. 1C, E and 2A are the same and derived from the same experiment (i.e., cell lysates) and blots were processed in parallel. (B) Kaplan–Meier analysis of RFS and OS of a curated cohort of TNBC patients divided into high and low *FADS1/FADS2* expressing using the best cutoff as described in the Materials and Methods section. HR and log-rank Mantel-Cox *P* values are shown. (C) *FADS1* and *FADS2* expression levels were analyzed in the different TNBC subtypes from the FUSCCTNBC cohort ($n = 82$ LAR, $n = 51$ MES, $n = 138$ BLIS, $n = 87$ IM). The LAR subtype was used as comparator in the statistical analysis. (D) Correlation analysis between the expression levels of *FADS1* and *FADS2* both in the LAR subtype (left) and in the whole cohort of TNBC of the FUSCCTNBC cohort (right). $R_s$ and log-rank Mantel-Cox *P* values are shown. (E) The ratio of PUFA/MUFA in 4T1 and 67NR cells for PC, LPC, PE, and LPE lipid classes was calculated as described in the Methods section ($n = 3$ biological replicates, see Dataset EV1). 67NR cells were used as comparator in the statistical analysis. (F) TNBC cells were analyzed by WB analysis using the antibodies described in the figure. (G) 24-h dose-response curve of RSL3 showed a differential effect between 67NR, 4T07, and 4T1 cells ($n = 3$ biological replicates). 67NR were used as comparator in the statistical analysis. (H) The GSH/GSSG concentration ratio was measured in 4T1 series' cells treated with 0.1 μM RSL3 for 24 h ($n = 3$ biological replicates). The untreated condition was used as comparator for each cell line in the statistical analysis. (I) Intracellular ROS levels were measured by DCFDA in 4T1 series' cells treated with 0.1 μM RSL3 for 24 h ($n = 3$ biological replicates). The untreated condition was used as comparator for each cell line in the statistical analysis. (J, K) Lipid peroxidation levels were evaluated by measuring the fluorescence intensity of BODIPY$^{581/591}$-C11 using confocal microscopy in murine 4T1 series' cells exposed to 1 μM RSL3 (J) or 5 μM erastin (K) for 2 h. Representative confocal images are shown. (Oxidized lipids: green; non-oxidized lipids: red; nuclei: blue, DAPI; scale bar, 10 μm). (L) In vivo response to RSL3 treatment in BALB/c mice ($n = 4$–6 mice/group). (M) Lipid peroxidation was evaluated by measuring the MDA intracellular accumulation in vehicle- and RSL3-treated tumors ($n = 4$ mice/group). (N) Total protein lysates from vehicle- and RSL3-treated tumors were analyzed by WB analysis using the antibody described in the figure. Relative quantification is shown using vehicle-treated tumors as comparator ($n = 3$ mice/group). (O) Representative 20× images of PLIN2 immunohistochemistry (IHC) staining and relative quantification (PLIN2 H-score) of four different areas derived from vehicle- and RSL3-treated 4T1 tumor specimens (scale bar, 200 μm). Relative quantification is shown using vehicle-treated tumors as comparator ($n = 5$ mice/group). Data information: In (C, E, G–I, L–O), data are presented as mean ± SEM. Statistical analysis was performed using one-way ANOVA followed by Dunnett's correction (C) or Tukey's correction (E, I), two-way ANOVA followed by Tukey's correction (H) or Bonferroni's correction (G, L), Two-tailed unpaired-sample Student *t*-test (M–O). Source data are available online for this figure.

levels (GSH/GSSG, Fig. 2H), (ii) ROS content using DCFDA probe (Fig. 2I) and (iii) lipid peroxidation using the fluorescent lipid peroxidation-sensitive dye BODIPY$^{581/591}$-C11 by confocal analyses (Fig. 2J,K) in the 4T1 series. Altered GSH/GSSG proportion and increased ROS content were observed in the 4T07 and 4T1 cells whereas no effects were observed in 67NR cells (Fig. 2H,I). Although BODIPY$^{581/591}$-C11 should be immediately oxidized after ferroptosis induction, this did not occur in 67NR exposed to RSL3. Conversely, a strong increase in the oxidized form of BODIPY$^{581/591}$-C11 was observed in the more aggressive cells exposed to RSL3 and erastin (Fig. 2J,K). Timing (2 h) and dosing (1 μM for RSL3 and 5 μM for erastin) were chosen not to induce cell death in cell lines.

To assess whether this metabolic vulnerability could be exploited in vivo, we used the syngeneic 4T1 model. 4T1 cells were subcutaneously injected in BALB/c mice where they formed palpable tumors after 7 days. Mice were then randomized and treated with RSL3 for 2 weeks. RSL3 significantly reduced tumor growth (Fig. 2L) without showing severe toxicity (Fig. EV2H). The ferroptosis execution in the RSL3-treated 4T1 model was confirmed by ex vivo analyses of the tumor-derived specimens that revealed increased levels of the final product of PUFA peroxidation malondialdehyde (MDA) in RSL3-treated when compared to vehicle-treated 4T1 tumors (Fig. 2M). Moreover, RSL3-treated 4T1 tumors showed reduced levels of total protein reduction, a trait that indicates the accumulation of oxidatively damaged proteins (Fig. 2N). Finally, IHC analysis of RSL3-treated 4T1 tumors showed reduced levels of LD content, as monitored by lower PLIN2 H-score expression (Fig. 2O, Bacci et al, 2024), hence reinforcing the interaction between deregulated lipid metabolism and ferroptosis susceptibility in metastatic TNBC.

To functionally confirm that aggressive TNBC cells undergo ferroptosis, we exposed the ferroptosis-sensitive 4T1, 4T07, D2A1-m1, and D2A1-m2 cells to RSL3 or erastin together with either the radical-trapping antioxidant ferrostatin-1 (Fer-1), the iron chelator agent deferoxamine (DFOM) or an ACSL4 inhibitor (ACSL4i, Fig. 3A). All these ferroptosis-preventing approaches were

sufficient to abrogate the cell death induced by both RSL3 (Fig. 3B–D) and erastin (Fig. EV3A,B) in metastatic murine cell lines while the less aggressive models were resistant to these agents (Appendix Fig. S1A–D).

To substantiate these observations, the analyses were expanded to include human TNBC models. Interestingly, the metastatic cells MDA-MB-468, MDA-MB-231, SUM159, and MDA-MB-453 showed increased susceptibility to cell death induced by RSL3 and/or erastin. Administration of Fer-1, DFOM, and ACSL4i effectively prevented the cell death-induced process (Figs. 3E and EV3C), hence confirming that human TNBC metastatic cells are ferroptosis-sensitive whereas no effect was reported in the less aggressive TNBC cells (i.e., BT20 and HCC1937, Appendix Fig. S1E–H).

## Targeting FADS1/2 prevents ferroptosis induction in aggressive TNBC

To demonstrate whether FADS1 and FADS2 have a role in the ferroptotic cell death in TNBC, we first exposed the ferroptosis-sensitive cells to SC-26196 (FADS2 inhibitor) and CP-24879 (FADS1/FADS2 dual inhibitor). The inhibition of FADS1 and FADS2 prevented the RSL3- (Fig. 4A–C) and the erastin-(Fig. EV4A–C) induced cell death in both aggressive murine and human TNBC cells. Conversely, the inhibition of SCD1 (using CAY10566) was not sufficient to prevent ferroptosis (Figs. 4A–C and EV4A–C). Confocal and flow-cytometry analyses of the oxidized form of BODIPY$^{581/591}$-C11 confirmed the prevention of lipid peroxidation exerted by FADS inhibition in RSL3- and erastin-treated 4T07 and 4T1 (Figs. 4D,E and EV4D), D2A1-m1 and D2A1-m2 (Figs. 4F and EV4E). Similarly, RSL3 and erastin rapidly elicited a strong increase in the peroxidation levels of MDA-MB-468, MDA-MB-231, SUM159, and MDA-MB-453 cells while both SC-26196 and CP-24879 prevented the drug-induced accumulation of toxic lipid peroxides (Figs. 4G,H and EV4F,G). Conversely, the less aggressive murine (67NR and D2A1) and

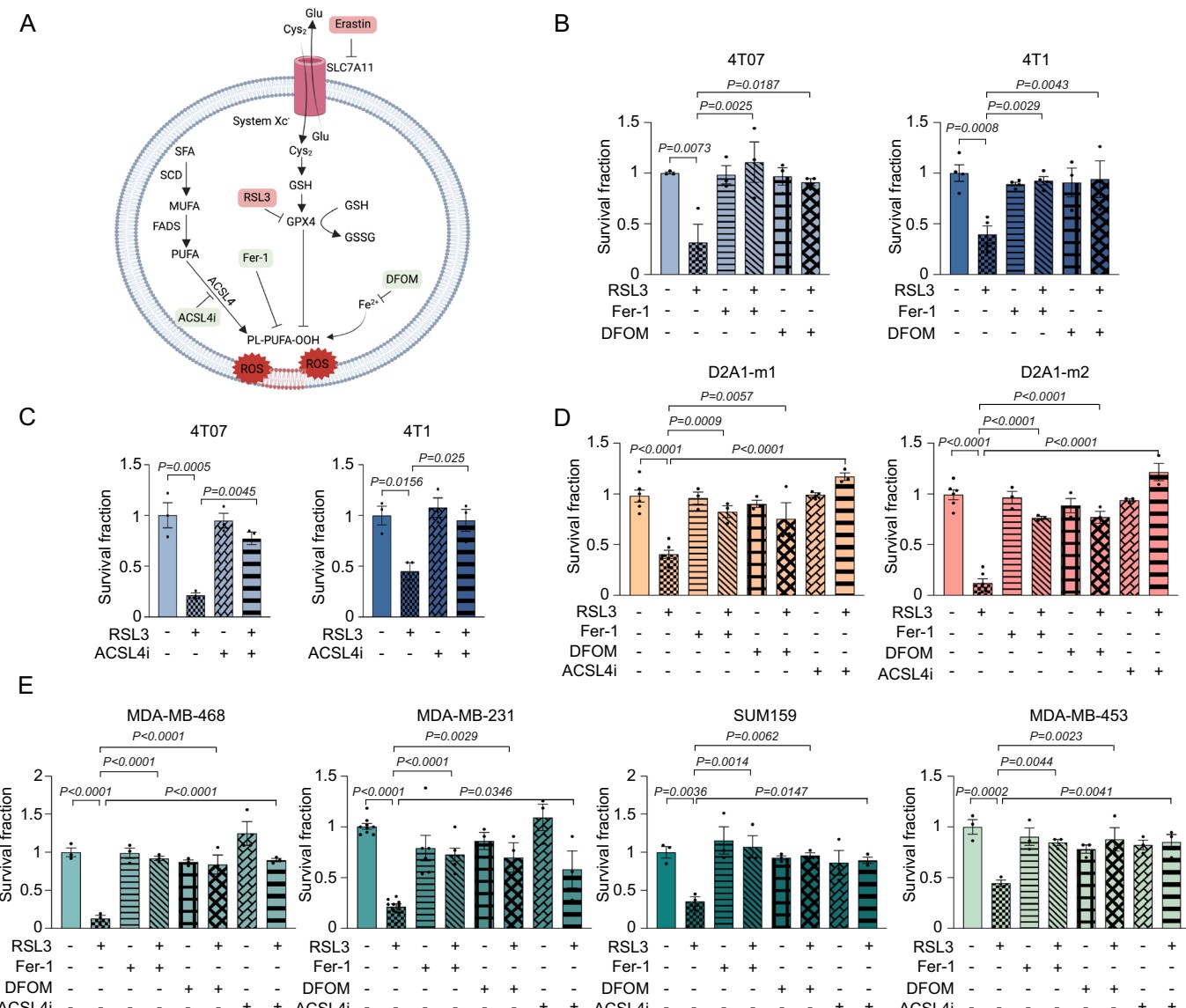

**Figure 3. Ferrostatin-1, Deferoxamine, and ACSL4 inhibition prevent RSL3-induced cell death in TNBC cells.**

(A) Schematic representation of ferroptosis execution. Glu: glutamate, Cys2: cystine. (B–D) 4T07, 4T1, D2A1-m1, and D2A1-m2 cells were pre-treated with 15 μM Fer-1 (B, D), 5 μM DFOM (B, D), or 10 μM ACSL4 inhibitor (ACSL4i, C, D) for 4 h, and then exposed overnight (ON, 16 h) to 0.1 μM (4T07 and 4T1) or 0.25 μM (D2A1-m1 and D2A1-m2) RSL3. After 24 h cells were subjected to cell viability assay (n = 3 biological replicates in either single or technical duplicate). The RSL3-treated cells were used as comparator in the statistical analysis. (E) Human metastatic MDA-MB-468, MDA-MB-231, SUM159, and MDA-MB-453 cells were pre-treated with 15 μM Fer-1, 5 μM DFOM, or 10 μM ACSL4i for 4 h, and then administrated ON with 0.25 μM RSL3. After 24 h cells were subjected to cell viability assay (n = 3 biological replicates in either single, duplicate or technical triplicate). The RSL3-treated cells were used as comparator in the statistical analysis. Data information: data are presented as mean ± SEM. Statistical analysis was performed using one-way ANOVA followed by Dunnett's correction (B, D, E) or Tukey's correction (C). Source data are available online for this figure.

human (BT20 and HCC1937) cellular models demonstrated a lack of responsiveness to both ferroptosis induction and FADS1/2 inhibition (Appendix Fig. S2A–H).

Moreover, since a recent study reported that anti-estrogen therapy sensitizes estrogen receptor-positive (ER+) breast cancer to ferroptosis (Liang et al, 2023), we explored whether endocrine therapy could induce ferroptosis by a mechanism involving FADS1/2 axis. Importantly, we showed that endocrine therapy (i.e., tamoxifen, TMX, fulvestrant, FULV, and estrogen deprivation,

-E2) sensitized ER + MCF7 and T47D cells to RSL3 while targeting FADS1/2 was not sufficient to avoid ferroptosis (Appendix Fig. S3A,B). Accordingly, when compared to a TNBC cell line (i.e., MDA-MB-453), MCF7 and T47D cells displayed lower levels of FADS1 and/or FADS2 (Appendix Fig. S3C), that were not affected by the administration of endocrine therapies (Appendix Fig. S3D).

Since no selective FADS1 inhibitors are available and CP-24879 is a FADS1/FADS2 dual inhibitor, we silenced FADS1 and FADS2

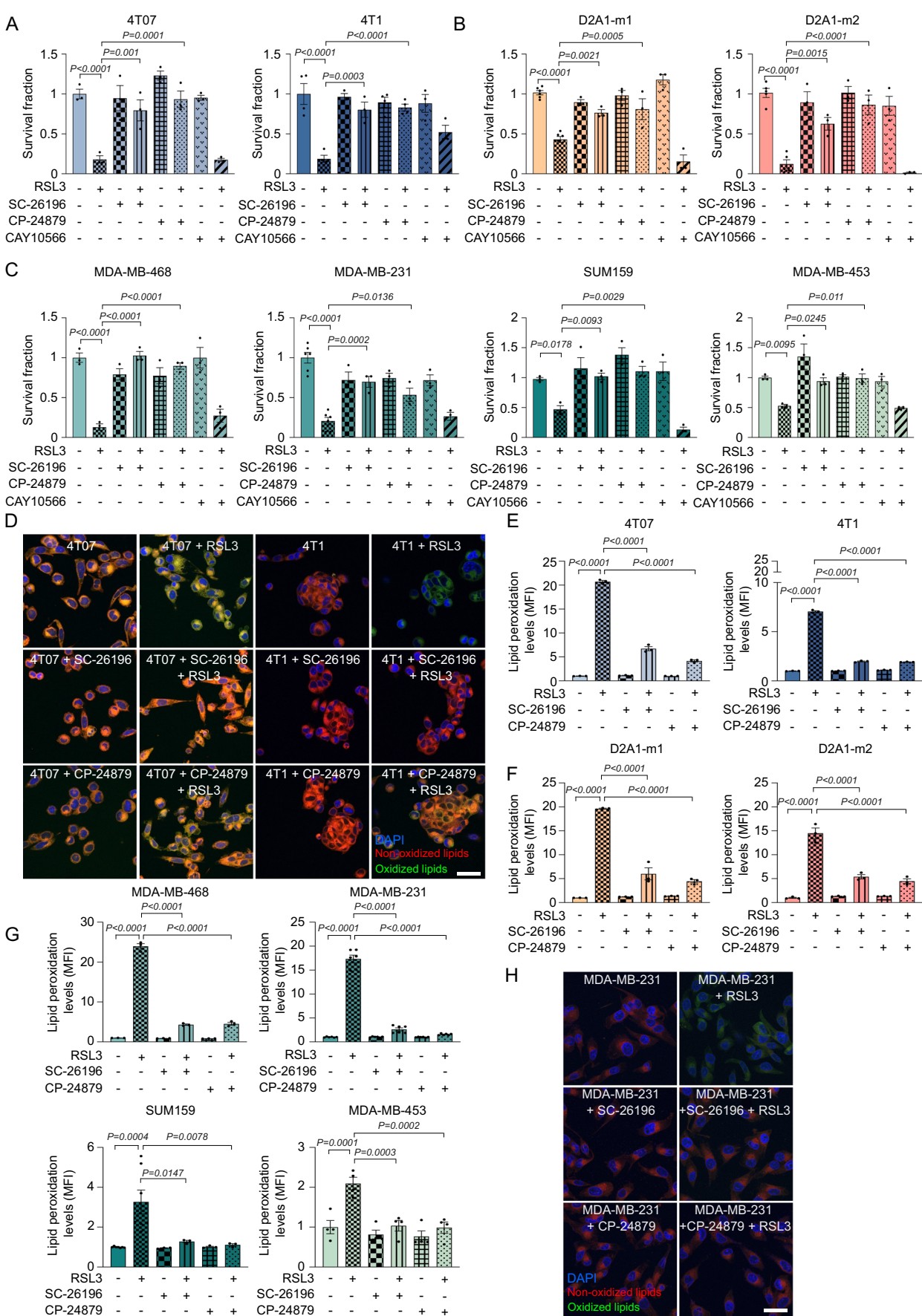

**Figure 4.  Targeting FADS1/2 prevents ferroptosis induction in aggressive TNBC.**

(A–C) TNBC metastatic murine 4T07 and 4T1 (A), D2A1-m1 and D2A1-m2 (B), and human MDA-MB-468, MDA-MB-231, SUM159, and MDA-MB-453 (C) cells were pre-treated with 10 µM FADS2 inhibitor (FADS2i, SC-26196), FADS1/2 inhibitor (FADS1/2i, CP-24879), and SCD1 inhibitor (SCD1i, CAY10566) for 4 h, exposed ON to 0.1 µM (4T1 series) or 0.25 µM (D2A1 series and human TNBC cells) RSL3, and subjected to cell viability assay ($n = 3$ biological replicates in either single or technical duplicate). The RSL3-treated cells were used as comparator in the statistical analysis. (D, E) 4T07 and 4T1 cells were pre-treated with 10 µM FADS2i or FADS1/2i for 4 h, exposed to 1 µM RSL3 for 2 h, and subjected to confocal analysis (D) and cytofluorimetric analysis (E) to measure lipid peroxidation. Representative pictures of BODIPY[581/591]-C11 stained cells are shown (oxidized lipids: green; non-oxidized lipids: red; nuclei: blue, DAPI; scale bar, 10 µm) ($n = 3$ biological replicates). The RSL3-treated cells were used as comparator in the statistical analysis. (F, G) Metastatic D2A1 (F) and human (G) TNBC cells were pre-treated with 10 µM FADS2i or FADS1/2i for 4 h, treated for 2 h with 1 µM RSL3, and subjected to cytofluorimetric analysis to measure lipid peroxidation ($n = 3$ biological replicates in either single, duplicate or technical triplicate). The RSL3-treated cells were used as comparator in the statistical analysis. (H) Highly metastatic MDA-MB-231 cells were pre-treated with 10 µM FADS2i or FADS1/2i for 4 h, treated for 2 h with 1 µM RSL3, and subjected to confocal analysis to measure lipid peroxidation. Representative pictures of BODIPY[581/591]-C11 stained cells are shown (oxidized lipids: green; non-oxidized lipids: red; nuclei: blue, DAPI; scale bar, 10 µm). Data information: In (A–C, E–G), data are presented as mean ± SEM. Statistical analysis was performed using one-way ANOVA followed by Dunnett's correction. Source data are available online for this figure.

in murine 4T1 and human MDA-MB-231 cells using 4 individual siRNAs pooled in 2 different combinations. As expected, FADS1 and FADS2 silencing in 4T1 (Fig. EV5A) and MDA-MB-231 cells (Fig. EV5E) prevented RSL3-induced ferroptosis cell death (Fig. EV5B,F) and lipid peroxidation analyzed by both confocal (Fig. EV5C,G) and FACS analyses (Fig. EV5D,H).

Moreover, to strengthen the role of FADS1/2 and validate the on-target effects of the FADS1/2 inhibitors and the FADS1 and FADS2 transient silencing, we performed a PiggyBac transposon-mediated transfer. This approach allowed the simultaneous introduction of 3 shRNA targeting FADS1 and 3 shRNA targeting FADS2 in 4T1 cells and the generation of stable *FADS1/FADS2* double knockdown transfected cells (FADS1/2[KD] 4T1, Fig. EV5I,J). To validate whether FADS1/2 have a crucial role in lipid intracellular homeostasis and metabolism, we subjected FADS1/2[KD] 4T1 and Scramble 4T1 cells to lipid extraction and subsequent nLC-MS/MS (Cattaneo et al, 2021). Lipidomic analysis identified 1003 lipid entities (Dataset EV2); of these 193 were over-represented and 52 under-represented in the FADS1/2[KD] 4T1 cells when compared to Scramble 4T1 cells (FDR < 0.05; adj. t-test; $P < 0.05$, fold-change = 2, Fig. 5A, Dataset EV2). Partial Least Square Discriminant Analysis (PLS-DA) of the identified entities showed a clear separation of the FADS1/2[KD] and Scramble 4T1 cells using the first two principal components with 58.2% and 17.8% of explained variance (Fig. 5B). Significantly, FADS1/2[KD] 4T1 cells exhibited a significant elevation in MUFA levels and a reduction in PUFA with double bonds ranging from 2 to 4 (Fig. 5C). Consequently, this alteration led to the decrease of the intracellular PUFA/MUFA ratio in the FADS1/2[KD] 4T1 cells (Fig. 5D). Functionally, FADS1/2[KD] 4T1 cells exhibited reduced sensitivity to ferroptosis induction upon exposure to concentrations of RSL3 and erastin that otherwise elicited cell death in FADS1/2 proficient cells, as validated by a dose-response analysis of the drugs (Fig. EV5K). Ectopic re-introduction of FADS1 and FADS2 in the FADS1/2[KD] 4T1 cells, by PiggyBac transposon-mediated transfer (FADS1/2[KD+OE]) (Fig. EV5I,J), restored RSL3 and erastin sensitivity (Fig. EV5K). Similarly, the PiggyBac-mediated introduction of FADS1 and FADS2 in 67NR (Fig. EV5L) sensitized the non-metastatic 67NR cells to ferroptosis (Fig. EV5M).

## Ferroptosis induction alters the lipidomic profile of sensitive TNBC cells but not that of the resistant cells

To explore the potential alterations in the intracellular lipid composition induced by ferroptosis execution in sensitive and

resistant cells, we subjected 67NR (resistant) and 4T1 (sensitive) cells, either untreated or exposed to a short-acute RSL3 treatment, to lipid extraction and subsequent lipidomic analysis (Cattaneo et al, 2021). PLS-DA of 1209 unambiguously identified lipids showed a clear separation between 4T1 cells and 4T1 exposed to RSL3 (Fig. 5E, Dataset EV1). Conversely, 67NR exposed or not to RSL3 did not cluster apart in the PLS-DA, suggesting very few potential differences in their lipidomic profile (Fig. 5E). Indeed, the analysis of lipid entities differentially expressed (adj. t-test; $P < 0.05$, fold-change = 1.5) upon RSL3 exposure identified 47 over-represented and 66 under-represented lipids in the 4T1 cells treated with RSL3 compared to untreated cells (Fig. 5F), whereas only 1 and 12 entities were over and under-represented in the 67NR pair (Fig. 5G), suggesting that RSL3 was neither able to induce cell death in the 67NR nor impact on their lipidomic profile.

## Exogenous FA supplementation alters ferroptosis execution in TNBC cells

In line with the major role of PUFA availability in sustaining ferroptosis execution in TNBC cells, we exposed the ferroptosis-resistant murine 67NR cells to exogenous FA supplementation. Adrenic acid (C22:4; *all-cis*-7,10,13,16-docosatetraenoic acid), erucic acid (C22:1; *cis*-13-docosenoic acid), and palmitoleic acid (C16:1; *cis*-9-hexadecenoic acid) were individually added to 67NR cell medium either alone or concomitantly to RSL3. In line with the hypothesis, the administration of the PUFA C22:4 sensitized 67NR to RSL3 exposure, as revealed by increased cell death (Fig. 6A) and lipid peroxidation (Fig. 6B). Conversely, MUFA (C22:1 and C16:1) administration did not sensitize 67NR to RSL3. Notably, the supplementation of the PUFA C22:4 enhanced RSL3- and erastin-induced cell death in the more aggressive 4T07, 4T1, and MDA-MB-231 cells while that of the MUFA 22:1 exerted a defensive role (Fig. 6C–F). These results reinforce the concept that PUFA availability is essential for ferroptosis execution, whereas MUFA could exert a protective role therefore indicating that PUFA/MUFA ratio alteration guides ferroptosis susceptibility in TNBC cells.

## Targeting LD biogenesis potentiates the effect of RSL3 in TNBC cells

LD have a crucial role in tumor progression and are involved in ferroptosis protection (Danielli et al, 2023). Particularly, storing PUFA into LD reduces the availability of double bonds for peroxidation (Dierge et al, 2021). Since 4T1 cells showed higher

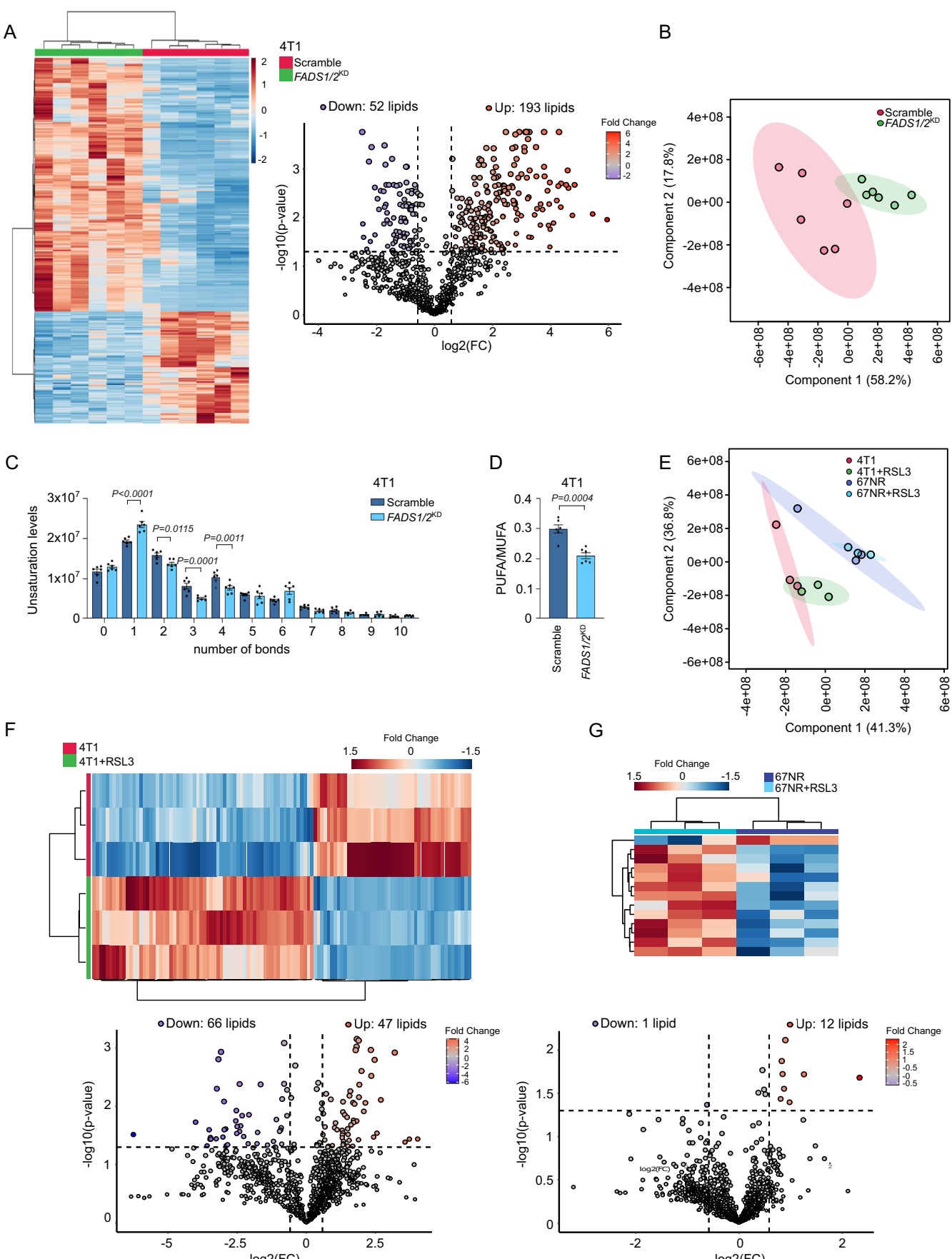

**Figure 5.** Both FADS1/2 knockdown and RSL3 administration alter the lipidomic profile of TNBC cells sensitive to ferroptosis induction.

(A) Unsupervised hierarchical clustering and heatmap of the significantly and differentially expressed lipid entities in FADS1/2$^{KD}$ 4T1 cells compared to Scramble 4T1 cells (left). The data were represented as volcano plot (right), highlighting the upregulated (red) and downregulated (blue) lipid species (fold change threshold of 1.5 and FDR < 0.05). (B) PLS-DA of the lipidomic profile of FADS1/2$^{KD}$ and Scramble 4T1 cells. (C) Unsaturation levels in FADS1/2$^{KD}$ and Scramble 4T1 cells ($n = 3$, in technical duplicates). (D) Ratio of PUFA/MUFA in FADS1/2$^{KD}$ and Scramble 4T1 cells ($n = 3$, in technical duplicates). (E) PLS-DA of the lipidomic profile of 67NR and 4T1 cells with and without 1 µM RSL3 for 2 h. (F) Unsupervised hierarchical clustering and heatmap of the significantly and differentially expressed lipid entities in 4T1 cells exposed to 1 µM RSL3 for 2 h (top). The data were represented as volcano plot (bottom), highlighting the upregulated (red) and downregulated (blue) lipid species (fold change threshold of 1.5 and $p$-value < 0.05). (G) Unsupervised hierarchical clustering and heatmap of the significantly and differentially expressed lipid entities in 67NR cells exposed to 1 µM RSL3 for 2 h (top). The data were represented as volcano plot (bottom), highlighting the upregulated (red) and downregulated (blue) lipid species (fold change threshold of 1.5 and $p$-value < 0.05). Data information: In (C, D), data are presented as mean ± SEM. Statistical analysis was performed using two-tailed unpaired-sample Student $t$-test (A, F, G) with FDR-adjusted analysis in (A), two-way ANOVA followed by Šidák's correction (C), or two-tailed unpaired-sample Student $t$-test (D). Source data are available online for this figure.

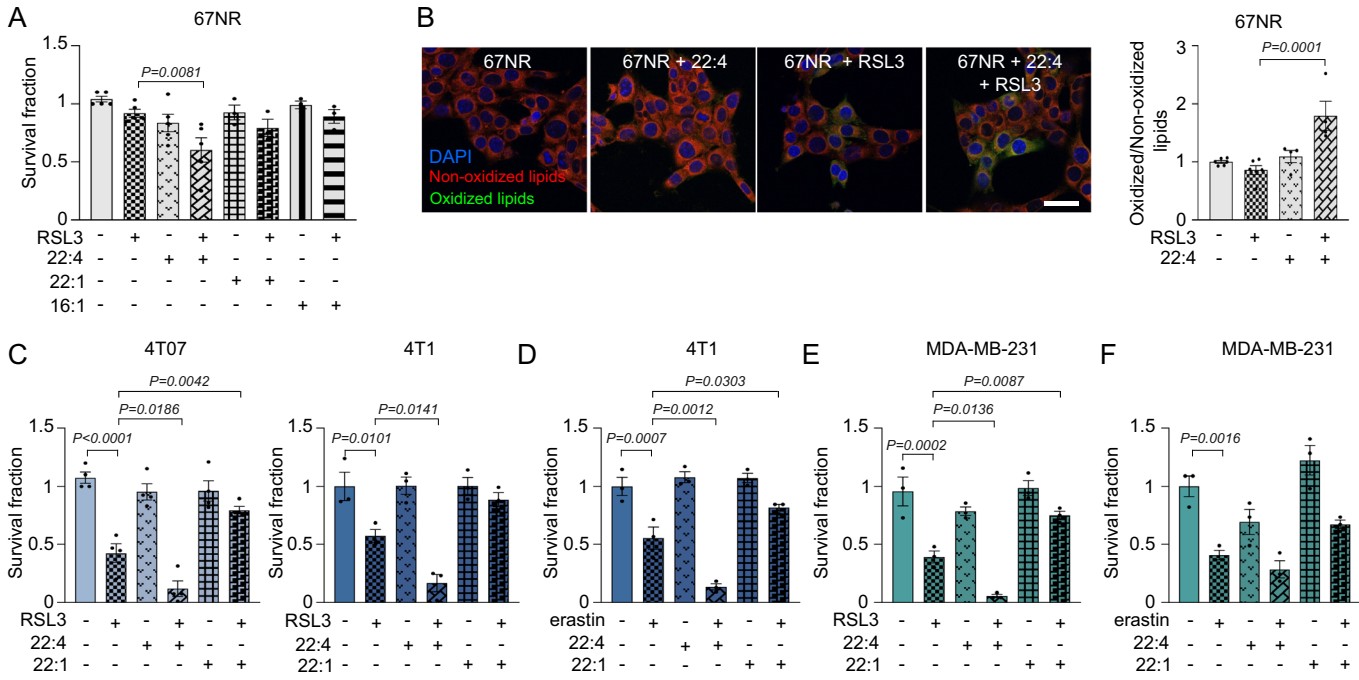

**Figure 6.** Exogenous FA supplementation alters ferroptosis susceptibility in TNBC cells.

(A) The non-metastatic 67NR cells were pre-treated with 10 µM C22:4, 10 µM C22:1, or 5 µM C16:1 for 4 h and then exposed ON to 0.1 µM RSL3. After 24 h cells were subjected to cell viability assay ($n = 3$ biological replicates in either single or technical duplicate). The RSL3-treated condition was used as comparator in the statistical analysis. (B) 67NR cells were pre-treated with 10 µM C22:4 for 4 h, then exposed for 2 h to 1 µM RSL3 and subjected to confocal analysis to measure lipid peroxidation. Representative pictures of BODIPY$^{581/591}$-C11 stained cells are shown (oxidized lipids: green; non-oxidized lipids: red; nuclei: blue, DAPI; scale bar, 10 µm). Quantification of BODIPY$^{581/591}$-C11 spots was reported ($n = 3$ biological replicates in either single or technical duplicate). The RSL3-treated condition was used as comparator in the statistical analysis. (C, D) TNBC aggressive 4T07 (C) and 4T1 (C, D) cells were pre-treated with 10 µM C22:4 or C22:1 for 4 h, exposed ON to 0.1 µM RSL3 (C) or 0.5 µM erastin (D) and subjected to cell viability assay ($n = 3$ biological replicates in either single or technical duplicate). The RSL3- or erastin-treated condition was used as comparator in the statistical analysis. (E, F) Human TNBC metastatic MDA-MB-231 cells were pre-treated with 10 µM C22:4 or C22:1 for 4 h, treated ON with 0.25 µM RSL3 (E) or 0.5 µM erastin (F), and subjected to cell viability assay ($n = 3$ biological replicates). The RSL3- or erastin-treated condition was used as comparator in the statistical analysis. Data information: data are presented as mean ± SEM. Statistical analysis was performed using one-way ANOVA followed by Dunnett's correction. Source data are available online for this figure.

LD content (Figs. 1D and EV1D) and higher FADS1/2 expression (Fig. 2A), we monitored LD content upon FADS1/2 inhibition. Interestingly, SC-26196 and CP-24879 administration reduced LD content, assessed using BODIPY$^{493/503}$ (Fig. 7A), suggesting that the increased LD content of 4T1 cells could be partially due to their ability to synthesize PUFA. Crucially, RSL3 administration resulted in a significant and rapid increase in the LD content of the more aggressive models (i.e., 4T07, 4T1, and MDA-MB-231), revealed by confocal (Fig. 7B) and/or cytofluorimetric analysis (Fig. 7C),

suggesting a potential scavenging mechanism exploited by the cells in an attempt to resist the ferroptosis-promoting insult. Conversely, no change was observed in the RSL3 insensitive 67NR (Fig. 7B, C). Although 4T07, 4T1, and MDA-MB-231 cells were already sensitive to RSL3 and erastin, impairing LD biogenesis using the DGAT1 and DGAT2 inhibitors suppressed the buffering capacity of LD and potentiated the cell death mediated effect induced by RSL3 or erastin (Fig. 7D,E), particularly in the more aggressive 4T1 and MDA-MB-231 cells. To note, in MDA-MB-231 only DGAT2

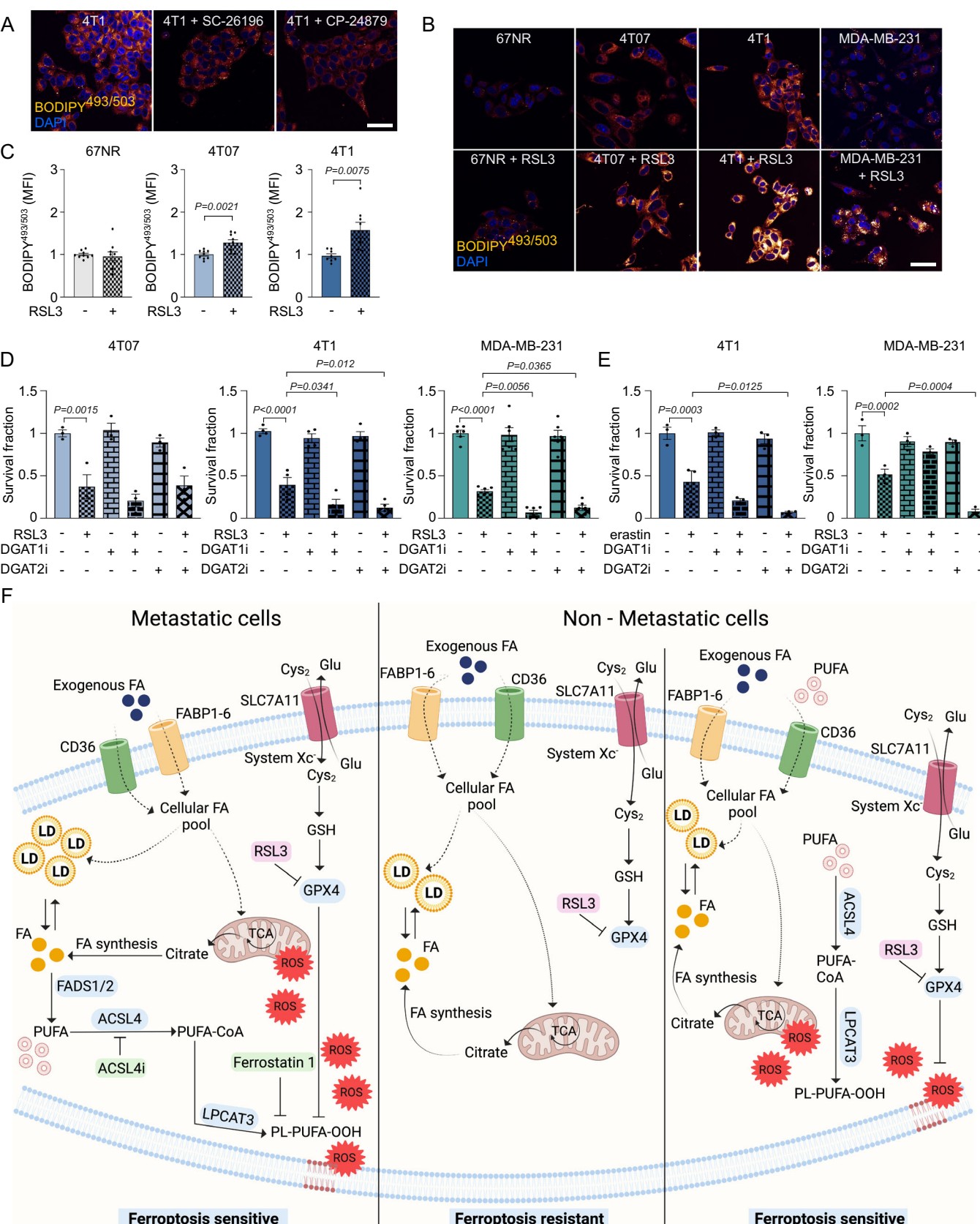

**Figure 7.  Targeting LD biogenesis potentiates the RSL3-mediated cell death in TNBC cells.**

(A) 4T1 cells treated for 24 h with 10 μM FADS2i or 10 μM FADS1/2i were subjected to BODIPY[493/503] staining by confocal microscopy. Representative images are shown (orange/yellow: LD; blue: DAPI, nuclei; scale bar, 10 μm). (B, C) TNBC murine 4T1 series' cells and human MDA-MB-231 cells treated with 1 μM RSL3 for 30 min were subjected to confocal (B) or cytofluorimetric (C) analyses. Representative pictures of BODIPY[493/503] stained cells are shown (orange/yellow: LD; blue: DAPI, nuclei; scale bar, 10 μm) ($n = 3$ biological replicates in either single, duplicate, or technical triplicate). (D) 4T07, 4T1, and MDA-MB-231 metastatic cells were pre-treated with 10 μM DGAT1i or DGAT2i for 4 h and then exposed ON to 0.1 μM (4T07 and 4T1) or 0.25 μM (MDA-MB-231) RSL3. After 24 h cells were subjected to cell viability assay ($n = 3$ biological replicates in either single or technical duplicate). The RSL3-treated condition was used as comparator in the statistical analysis. (E) 4T1 and MDA-MB-231 metastatic cells were pre-treated with 10 μM DGAT1i or DGAT2i for 4 h and then exposed ON to 0.5 μM erastin. After 24 h cells were subjected to cell viability assay ($n = 3$ biological replicates). The erastin-treated condition was used as comparator in the statistical analysis. Data information: In (C–E), data are presented as mean ± SEM. Statistical analysis was performed using two-tailed unpaired-sample Student t-test (C), one-way ANOVA followed by Dunnett's correction (D, E). (F) Proposed mechanism of ferroptosis susceptibility in TNBC and the underlying metabolic crosstalk between lipid upload, usage, and metabolism and the ROS-detoxifying system. Source data are available online for this figure.

potentiated the erastin-mediated cell death (Fig. 7E), while in 4T07 cells a modest increase in RSL3 induced cell death was observed only when co-administrated with DGAT1 inhibitor but not with the DGAT2 inhibitor (Fig. 7D). Collectively, these data propose a novel synthetic lethality therapeutic approach that could be further investigated in those ferroptosis-resistant cells that are characterized by increased LD content (Fig. 7F).

## Discussion

This study identifies a novel lipid metabolic reprogramming as a potential metabolic vulnerability that can be exploited in a subset of aggressive TNBC. Indeed, we have shown that TNBC characterized by increased FADS1/2 expression are susceptible to ferroptosis-inducing agents. This is crucial in addressing a biological and clinical need as it may allow stratifying patients that could benefit from a tailored approach in a subset of breast cancers in which therapy remains a challenge.

A recent integrated multi-omics analysis of a large cohort of TNBC reported heterogeneous phenotypes in ferroptosis-related metabolites and metabolic pathways (Yang et al, 2023). Particularly, the LAR TNBC subtype is characterized by ROS accumulation, increased levels of oxidized phosphatidylethanolamines, and deregulation of glutathione metabolism, which render LAR TNBC sensitive to GPX4 inhibitors (Yang et al, 2023). Although the authors reported increased levels of intracellular PUFA in ferroptosis-sensitive LAR TNBC, they did not prove that this was due to increased PUFA biosynthesis (i.e., FADS1/2 activity). However, in other cancer types, the increased expression levels and subsequent activity of the enzymatic asset involved in the formation of long and unsaturated FA is a major determinant of ferroptosis sensitivity. Lee et al reported that mesenchymal-type gastric cancers, but not those of the intestinal subtype, are sensitive to the GPX4 inhibitors RSL3 and ML-210 (Lee et al, 2020). An in-depth metabolic and metabolomic profiling of this ferroptosis-sensitive subset revealed increased intracellular levels of PUFA, particularly PE-linked arachidonic acid and adrenic acid, which are synthesized in cancers with high expression levels of the ELOVL5 (elongation of very long chain fatty acids protein 5) and FADS1 (Lee et al, 2020). Moreover, Xuan et al reported increased levels and activity of SCD1 and FADS2 in metastatic ovarian cancers (Xuan et al, 2022).

However, the intracellular levels of oxidable PUFA in ferroptosis-sensitive cells are also dependent on extracellular availability and exposure to exogenous FA, since it is established that altering the intracellular relative composition between MUFA and PUFA by exogenous FA supplementation is essential for ferroptosis execution (Magtanong et al, 2019). Indeed, Ubellacker et al showed an anti-ferroptosis effect of oleic acid supplementation in melanoma (Ubellacker et al, 2020) and Dierge et al showed that cancer cells can undergo ferroptosis in an acidic environment when supplemented with PUFA but not with oleic acid (Dierge et al, 2021). Mechanistically, the authors showed that preventing LD formation upon PUFA supplementation did not protect cancer cells from the enhanced uptake of PUFA and subsequent lipid peroxidation, particularly of those PUFA incorporated into the plasma membrane. In line with these reports, we herein demonstrated that PUFA administration sensitized the less aggressive TNBC cells to ferroptosis and potentiated the effects of the ferroptosis-inducing agents in the aggressive models, whereas MUFA supplementation had the opposite effects. Consistent with these findings, reintroducing FADS1/2 into metastatic TNBC cells, previously subjected to FADS1/2 downregulation, effectively reinstated the cellular phenotype and susceptibility to ferroptosis. Given the lipidomic analysis indicating elevated levels of MUFA and reduced levels of PUFA in FADS1/2 knockdown cells, it is reasonable to suggest that the outcomes following FADS1/2 reinsertion may be dependent on the alteration of the PUFA/MUFA ratio.

The importance of the crosstalk between LD and ferroptosis susceptibility emerged also in our study (Fig. 7F). Indeed, when aggressive TNBC cells are exposed to ferroptosis-inducing agents, they immediately activate a "defensive" mechanism involving a rapid LD formation. However, our data showed that aggressive TNBC cells are characterized by high basal levels of LD, a feature that is paralleled by increased ROS levels and lipid biosynthesis. It is reasonable to postulate that, although LD turnover can be activated in these cells, it is not sufficient to buffer the deleterious effects induced by RSL3 that concomitantly influence lipid availability and redox homeostasis. An important aspect that was not addressed by our study is why increasing desaturase expression might be advantageous for aggressive TNBC.

Several mechanisms may underlie the observed association between increased PUFA levels and enhanced invasive potential in our cancer models. Firstly, an enrichment of PUFA within the plasma membrane could directly facilitate cancer cell movement. Indeed, it has been demonstrated that enhanced PUFA content promotes membrane fluidity, potentially increasing the efficiency of cellular protrusions and migration (Harayama and Antonny, 2023). This phenomenon might be relevant to the invasive behavior observed in our models. Secondly, the observed PUFA increase could be linked to the production of pro-tumorigenic eicosanoids.

Eicosanoids are a class of bioactive signaling lipids derived from arachidonic acid, a PUFA itself. Seminal studies have established the crucial role of eicosanoids in promoting tumor growth and progression through various signaling pathways (Koundouros et al, 2020; Wang and Dubois, 2010). Finally, elevated PUFA levels, regardless of their source (i.e., dietary intake or produced by cancer cells themselves), could contribute to the establishment of a pro-inflammatory tumor microenvironment. This inflammatory *milieu* is well-documented to promote tumor initiation and progression (Marion-Letellier et al, 2015; Vriens et al, 2019).

In conclusion, our data show that lipid metabolism can reveal metabolic vulnerabilities within a subset of breast cancers that lack prognostic markers and selective therapeutic strategies. Importantly, many preclinical studies have proposed to target lipid metabolism, but the clinical translation has been so far challenging and ineffective. Ferroptosis-inducing agents have been proposed in different tumor contexts although so far this approach has been tested in preclinical models. However, dietary approaches that influence iron availability (Delesderrier et al, 2023) and selenocysteine synthesis (Chen et al, 2013), two major players involved in ferroptosis execution, could have a more rapid therapeutic translation and, therefore, identifying those TNBC patients that are ferroptosis sensitive (or resistant) may be of clinical relevance. Finally, since FA supplementation alters cancer ferroptosis sensitivity, dietary approaches will be essential for potentiating the efficacy of a pro-ferroptosis approach not only in ferroptosis-sensitive tumors but also in those that are intrinsically resistant. In addition, since ferroptosis execution is immunogenic (Yang et al, 2023), ferroptosis-promoting agents can be investigated prospectively not only as monotherapy but in combination with immunotherapy or other anti-tumoral drugs (Minami et al, 2023). Impacting on the cancer desaturase activity (whether expressed by the cancer cells or by other tumor-associated stromal cells) could therefore change the FA composition of the tumor microenvironment and influence the success of the ferroptosis-inducing approach.

# Methods

## Cell cultures and reagents

Human (BT20, HCC1937, MDA-MB-231, MDA-MB-468, MDA-MB-453, SUM159) and murine (67NR, 4T07, 4T1, D2A1, D2A1-m1, D2A1-m2) female breast cancer cells were obtained from Prof. Clare Isacke (ICR, London, UK) and maintained at 37 °C/5% $CO_2$ in phenol red Dulbecco's Modified Eagle's Medium (DMEM, Euroclone #ECB7501L) supplemented with 10% Fetal Bovine Serum (FBS, Euroclone #ECS0180L), 2 mmol/L glutaMAX (Gibco #35050-061), and 1% penicillin/streptomycin (Sigma #P4333). Cells were short tandem repeat tested, amplified, stocked, routinely subjected to mycoplasma testing, and once thawed were kept in culture for a maximum of 3-4 weeks. SB-204990 (i.e., ACLY inhibitor, (3R,5S)-rel-5-[6-(2,4-dichlorophenyl)-hexyl]-tetrahydro-3-hydroxy-2-oxo-3-furanacetic acid, #HY-16450), RSL3 ((1S,3R)-2-(2-chloroacetyl)-2,3,4,9-tetrahydro-1-[4-(methoxycarbonyl)phenyl]-1H-pyrido[3,4-b]indole-3-carboxylic acid, methyl ester, #HY-100218A), erastin (2-[1-[4-[2-(4-chlorophenoxy)acetyl]-1-piperazinyl]ethyl]-3-(2-ethoxyphenyl)-4(3H)-quinazolinone, #HY-15763), A922500 (i.e., DGAT1 inhibitor, DGAT1i, (1R,2R)-2-[[4'-[[Phenylamino)carbonyl]amino]    [1,1'-biphenyl]-4-yl]carbonyl] cyclopentanecarboxylic acid, #252801), PF-06424439 (i.e., DGAT2 inhibitor, DGAT2i, [(3R)-1-[2-[1-(4-chloro-1H-pyrazol-1-yl)cyclopropyl]-3H-imidazo[4,5-b]pyridin-5-yl]-3-piperidinyl]-1-pyrrolidinyl-methanone, monomethanesulfonate, #PZ0233), CP-24879 (i.e., FADS1/2 inhibitor, 4-(3-methylbutoxy)-benzenamine, monohydrochloride, #C9115), and ATGListatin (i.e., ATGL inhibitor, N'-[4'-(dimethylamino)[1,1'-biphenyl]-3-yl]-N,N-dimethyl-urea, #HY-15859) were purchased from MedChemExpress, dissolved in DMSO and used at the indicated concentrations as described in the Figure Legends. TVB3166 (i.e., FASN inhibitor, 4-{1-[5-(4,5-dimethyl-2H-pyrazol-3-yl)-2,4-dimethyl-benzoyl]-azetidin-3-yl}-benzonitrile, #SML1694), SC-26196 (i.e., FADS2 inhibitor, α,α-Diphenyl-4-[(3-pyridinylmethylene)amino]-1-piperazinepentane-nitrile, #PZ0176), CAY10566 (i.e., SCD1 inhibitor, 2R-[[4'-[[(phenylamino)carbonyl]amino][1R,1'-biphenyl]-4-yl-carbonyl]-cyclopentanecarboxylic acid, #SML2980), etomoxir (i.e., FAO inhibitor, 2-[6-(4-chlorophenoxy)-hexyl]-oxirane-2-carboxylate, #E1905), ferrostatin-1 (Fer-1, ethyl 3-amino-4-(cyclohexylamino) benzoate, #SML0583), deferoxamine mesylate salt (i.e., iron chelator, DFOM, butanediamide, N'-5-[[4-[[5-(acetylhydroxyamino)pentyl]amino]-1,4dioxobutyl]hydroxyamino]pentyl]-N-(5-aminopentyl)-N-hydroxy-, monomethanesulfonate, #D9533), and rosiglitazione (i.e., ACSL4 inhibitor, ACSL4i, 5-[[4-[2-(Methyl-2-pyridinylamino)ethoxy]phenyl]methyl]-2,4-thiazolidinedione, #R2408) from Sigma-Aldrich were also dissolved in DMSO. TOFA (i.e., ACC1 inhibitor, 5-(tetradecyloxy)-2-furoic acid, #sc-200653) was purchased from Santa Cruz Biotechnology and dissolved in DMSO. The exogenously administrated FA adrenic acid (C22:4; *all-cis*-7,10,13,16-docosatetraenoic acid, #D3659), erucic acid (C22:1; cis-13-docosenoic acid, #E3385), and palmitoleic acid (C16:1; *cis*-9-hexadecenoic acid, #P9417) were purchased from Sigma-Aldrich and dissolved in ethanol.

## Mouse models and in vivo experiments

6-week-old female BALB/c mice were orthotopically inoculated with 4T1 ($2.5 \times 10^5$ cells, $n = 6$) resuspended in 50 μL of Dulbecco's phosphate buffered saline (PBS, Euroclone, #ECB4053) in a single flank injection. Tumors were grown to ~4.5 mm diameter (day 8, mean value) and then mice were randomly assigned to the vehicle or the RSL3-treated group. RSL3 (40 mg/kg) was injected intraperitoneally every other day in 50 μL of a mixture of 40% PEG300, 45% saline, 10% DMSO, and 5% Tween-80 starting from day 8. Tumor volume was monitored from day 7 onward by caliper measurements of the two largest tumor diameters. Volumes were calculated according to the formula: $a \times b^2 \times \pi/6$, where $a$ and $b$ are orthogonal tumor diameters. Animal weight was measured three times a week. Animals were culled on day 23 when tumor volumes of the vehicle-treated group reached the maximum tumor size. The number of animals was chosen to ensure adequate statistical power and was based on previous experience with the 4T1 model. Mice were maintained in specific pathogen-free conditions (Allentown Caging Equipment, Allentown Inc.) and treated in accordance with University Ethical Committee and European guidelines under directive 2010/63. Animal experimentation was carried out under the approval of the Italian Ministry of Health (authorization #741/2022).

## Survival assay

Human and murine breast cancer cells were seeded into 12-well plates at $3$–$5 \times 10^4$ cells/well in either standard conditions (see Cell cultures and reagents) or experimental conditions such as RNA interfering (i.e., siFADS1, siFADS2, FADS1/FADS2 double knockdown, FADS1/2$^{KD}$, FADS1/2 overexpression, FADS1/2$^{OE}$) or drug administration (e.g., 0.1–0.25 μM of the GPX4 inhibitor, RSL3, or 0.5 μM of the SLC7A11 inhibitor, erastin), as described in the Figures and relative legends. Unless specified otherwise, 24 h post-procedure cells were extensively washed with PBS, fixed with 4% formaldehyde, and subjected to the colorimetric evaluation of cell proliferation and viability using crystal violet (triphenylmethane dye 4-[(4-dimethylaminophenyl)-phenyl-methyl]-*N*, *N*-dimethyl-alanine, Sigma-Aldrich #548-62-9), thus dried overnight (ON, 16 h). The incorporated amount of crystal violet within the adherent living cells was solubilized with a maximum of 500 μl/well of 2% SDS (Sodium Dodecyl Sulfate) and quantified by measuring the absorbance at 595 nm using a microplate reader. Alternatively, cell counts were performed at least in triplicate by three analysts under a 10x objective according to the standard methodology.

## Transient siRNA transfection

4T1 and MDA-MB-231 cells were seeded into 6-well plates ($3 \times 10^5$ per well) to achieve 70% confluence the following day, when cells were transfected with either 100 nmol/L siRNA targeting FADS1 or FADS2 (siFADS1, siFADS2, GE Healthcare Dharmacon), used as different combination pools of 4 individual siRNAs (siFADS1_pool1 and pool2, siFADS2_pool1 and pool2) or the respective negative control (non-targeting small interfering RNA, siCTR, GE Healthcare Dharmacon) using Lipofectamine RNAiMAX Reagent (Thermo Fisher Scientific #13778-150) and Opti-MEM (GIBCO #31985062) accordingly to manufacturer's instructions. The functional analyses were performed 24 h after transfection as described in the Figure Legends.

## PiggyBac transposon-mediated shRNA transfer and generation of stably transfected cells

4T1 cells were seeded into 6-well plates ($1.5 \times 10^5$ per well) to achieve a 40–50% confluence the following day when cells were transfected with the PiggyBac Vector System to generate a stable FADS1/2 knockdown cell line. 4T1 cells were transfected with either 5 μg of piggyBac-shFADS1 Neo and 5 μg of piggyBac-shFADS2 Neo (shFADS1 and shFADS2, Vector Builder) or the scrambled control (Scramble, Vector Builder) together with 200 ng of pBase using 5 μg of X-tremeGENE HP DNA Transfection Reagent (Roche #06366546001) according to manufacturer's instructions. For the add-back control (i.e., rescue experimental condition), FADS1/2 double knockdown 4T1 cells were seeded into 6-well plates ($1.5 \times 10^5$ per well 24 h prior to transfection) and subjected to transfection with 5 μg of piggyBac-FADS1:FADS2 Neo and 200 ng of pBase using 5 μg of X-tremeGENE HP DNA Transfection Reagent according to the manufacturer's instructions. Similarly, to generate a stable cell line expressing FADS1 and FADS2, 67NR cells were seeded into 6-well plates ($2 \times 10^5$ per well) and transfected with the same concentrations of the piggyBac-

FADS1:FADS2 Neo and pBase vectors using X-tremeGENE HP DNA Transfection Reagent. After transfection, cells were selected with G418 (Gibco #11811-031) and Puromycin (Sigma-Aldrich #P8833) for 48 h before validating the knockdown efficiency by qRT-PCR and/or western blot analyses and then subjecting the stable generated cell lines to functional assays.

## Western blotting analysis

Human and murine breast cancer cells were washed with PBS and lysed on ice using 1x Laemmli Sample Buffer (Biorad #161-0737) supplemented with protease (Sigma-Aldrich #P8340) and phosphatase inhibitors (#P0044). Protein concentrations were measured using BCA assay (Sigma-Aldrich #1003290033). 40–50 μg of cell lysate were loaded in precast SDS-PAGE (sodium dodecyl sulfate–polyacrylamide gel electrophoresis) gels (Biorad #456-8096) and then transferred onto nitrocellulose membrane by Trans-Blot Turbo Transfer Pack (Biorad #170-4157). The immunoblots were incubated in PBS-T (PBS with 0.05% tween-20) containing 5% non-fat dry milk at room temperature for 1 h and then probed with primary and appropriate secondary antibodies. The following antibodies were used: ACLY (Santa Cruz Biotechnology #sc-517267), FASN (Santa Cruz Biotechnology #sc-55580), PLIN2 (Cell Signaling Technology #45535), FADS1 (Sigma-Aldrich #SAB2100744), FADS2 (Sigma-Aldrich #SAB1303849), SCD1 (Santa Cruz Biotechnology #sc-515844), SCD2 (Santa Cruz Biotechnology #sc-518034), ACSL4 (Santa Cruz Biotechnology #sc-365230), ACSL5 (Santa Cruz Biotechnology #sc-365478), HRP (horseradish peroxidase)-Conjugated Streptavidin (Thermo Fisher Scientific #N100), HSP90 (Santa Cruz Biotechnology #sc-69703), and β-ACTIN (Cell Signaling #3700), all diluted 1:1000 in PBS-T containing 5% non-fat dry milk. The loading controls used in the WB analyses were derived from the same experiment (i.e., cell lysates) and blots were processed in parallel when it was not possible to run on the same gel. The loading controls of the human cell lines (β-ACTIN and HSP90) in Figs. 1C,E, 2A are the same.

## RNA extraction and quantitative real-time PCR (qRT-PCR) analysis

Total RNA was extracted using RNeasy plus kit (QIAGEN #74134), quantified using Nanodrop 1000 (Thermo Fisher Scientific), and 500 ng were reverse transcribed using the iScript gDNA Clear cDNA Synthesis Kit (Biorad #172-5035). qRT-PCR was performed using the CFX96 Touch Real-Time PCR Detection System (Biorad) using TaqMan Universal PCR Master Mix (Thermo Fisher Scientific #4305719). The following probes were used: ACLY, FASN, PLIN2, FADS1, FADS2, and ACSL4 (Thermo Fisher Scientific). Data were normalized on TBP (TATA-Box Binding Protein) or GAPDH (Thermo Fisher Scientific). The relative quantity was determined using ΔΔCt by the CFX Maestro software (BioRad).

## Radioactive assay

Breast cancer cells ($8 \times 10^4$ cells/well) were seeded into 12-well plates. To analyze the incorporation of $^{14}$C-glucose, $^{14}$C-lactate, $^{14}$C-acetate, and $^{14}$C-glutamine into lipids, culture media were supplemented overnight (ON, 16 h) with 1 mCi of each

radiolabeled metabolite (Perkin Elmer #NEC042V250UC, #NEC599050UC, #NEC553250UC, #NEC451050UC). Cells were then washed 3 times in ice-cold PBS and lysed in RIPA buffer (Thermo Fisher Scientific #89900). Samples were first resuspended in 4 volumes of a CHCl3:MeOH (1:1) solution and then an additional volume of $dH_2O$ was added. The solution was then centrifuged at 1000 rpm for 5–10 min at room temperature. The lower phase (i.e., lipids) was collected, transferred to a scintillation vial, and the incorporated radioactive metabolite-derived signal was measured on the scintillation counter and normalized to the total protein content.

## Seahorse-based oxidative phenotyping

67NR, 4T07, and 4T1 cells were seeded in XFe96 cell culture plates with $3 \times 10^4$ cells per well (10 technical replicates) in 80 µL of culture medium, subjected to the experimental conditions described in the Figures, and incubated at 37 °C. 24 h post seeding, the culture medium was replaced with 180 µL of XF DMEM supplemented with 25 mM glucose (Sigma #G8644) and 2 mM glutaMAX. Cells were incubated for 1 h at 37 °C in a non-CO2 incubator to allow them to pre-equilibrate with the XF DMEM. The oxygen consumption rate (OCR) was quantified using the Seahorse Extracellular Flux Analyzer (XFe96, Agilent Technologies). An accurate titration with the uncoupler FCCP was performed for each cell type. The addition of the ATP synthase inhibitor oligomycin (1.5 µM), the proton uncoupler FCCP (1 µM), the respiratory complex I inhibitor rotenone (0.5 µM), and the respiratory complex III inhibitor antimycin A (0.5 µM) was carried out at the times indicated. This assay measures the cellular substrate oxidation by assessing changes in the OCR of live cells when the entry of specific mitochondria (i.e., TCA cycle) fueling substrates is impaired. Indeed, the XF Cell Mito Stress Test was combined with substrate pathway-specific inhibitors: etomoxir (4 µM) for long-chain fatty acids (LCFA) through inhibition of carnitine palmitoyl transferase 1a (CPT1a), UK5099 (2 µM) for glucose and/or pyruvate through inhibition of the mitochondrial pyruvate carrier (MPC), and BPTES (3 µM) for inhibition of glutamine through glutaminase 1 (GLS-1). Protein quantification was used to normalize the results. Basal respiration is calculated as the last rate measurement before oligomycin injection − non-mitochondrial respiration rate. Maximal respiration is calculated as the maximum rate measurement after FCCP injection – non-mitochondrial respiration rate.

## Oroboros O2k-FluoRespirometer

Oxygen consumption was analyzed in 2 mL glass chambers at 37 °C using the Oroboros Oxygraph-2K high-resolution respirometer (Oroboros Instruments, Innsbruck, Austria) and the substrate, uncoupler, inhibitor, titration (SUIT) protocol (D009) (Ye and Hoppel, 2013). The oxygen flux normalized on the cell number is calculated as the negative time derivative of the oxygen concentration, measured in sealed chambers, and normalized on the instrumental background (measured in a dedicated experiment before assaying the cells). Murine breast cancer cells cultured in basal condition and treated with 40 µM Etomoxir or vehicle (DMSO) for 30 min were subjected to respirometry analysis. After instrumental air calibration, $5–7 \times 10^5$ cells resuspended in complete culture medium were introduced into the chambers and

the basal respiratory activity was measured as routine respiration (R). The LEAK state (L) represents the non-phosphorylating state of uncoupled respiration due to proton leak, proton and electron slip, and cation cycling (Pesta and Gnaiger, 2012) after the inhibition of ATP synthase by oligomycin administration (5 nM). The capability of the electron transfer system (ETS) was measured by uncoupler titrations using the uncoupler Carbonyl Cyanide 3-ChloroPhenylhydrazone (CCCP; 1.5 µM/titration steps) as the readout of the maximal capacity of oxygen utilization (E). The residual oxygen consumption (ROX) that remains after the inhibition of ETS was determined by antimycin A injection (2.5 µM). Data acquisition and analysis were performed using DatLab software (Oroboros Instrument, Innsbruck, Austria) and the oxygen fluxes recorded in the individual titration steps were corrected for ROX.

## Lipidomic analysis

Lipids were extracted using a single-step extraction protocol with methanol and chloroform as described previously (Cattaneo et al, 2021). Briefly, cell pellets (equivalent to $1–3 \times 10^3$ cells) were resuspended in 200 µL of MilliQ water and mechanically disrupted by passing 20 times through a 26 G syringe needle. Proteins were extracted from 20 µL lysate by adding 5 µL of lysis buffer (10% NP40, 2% SDS in PBS) and quantified by BCA protein assay kit. Lipids were extracted starting from equivalent lysate corresponding to 50 µg of proteins. Lysates were made up to 170 µL with cold water and spiked in with 1 µL of SPLASH® LIPIDOMIX® Mass Spec Standard. Lipid extraction was performed by adding 700 µL of methanol and subsequently 350 µL of chloroform. Samples were mixed on the orbital shaker for 15 min at 4 °C. After that, 350 µL of water/chloroform (1:1 v/v) were added to each suspension and centrifuged at $10,000 \times g$ for 10 min at 4 °C. The organic phase was recovered, dried out, and finally resuspended in 25 µL of a buffer composed by 90% of Lipidomics Buffer A (95% of mobile phase A = ACN:$H_2O$ 40:60; 5 mM $NH_4COOCH_3$; 0.1% HCOOH and 5% of mobile phase B = Isopropanol:$H_2O$ 90:10; 5 mM $NH_4COOCH_3$; 0.1% HCOOH) and 10% of ethanol. 1 µL was injected on the nLC Ekspert LC400 set in nano configuration coupled with the mass spectrometer Triple TOF 6600 (AB Sciex). Chromatography was performed using an in-house packed nanocolumn Kinetex EVO C18, 1.7 µm, 100 A (Phenomenex, Torrance, CA, USA), $0.75 \times 100$ mm at room temperature. The gradient started from 5% of mobile phase B and was linearly increased to 100% B in 5 min, maintained for 45 min, then returned to the initial ratio in 2 min and maintained for 8 min at a flow rate of 150 nL/min. Samples were analyzed in positive mode with electrospray ionization. Spectra were acquired by full-mass scan from 200 to 1700 $m/z$ and information-dependent acquisition (IDA) from 50 to 1800 $m/z$ (top 8 spectra per cycle). The de-clustering potential was fixed at 80 eV, and collision energy at 40 eV, target ions were excluded for 20 s after 2 occurrences. Wiff files were processed using the open-source software MS-DIAL version 4.8 with manual inspection of peak integrations. Peak areas were normalized by using the sum of lipid species areas for each sample. For lipid unsaturation analysis, normalized lipid areas were grouped into lipid classes and further sub-grouped according to the number of double bonds. For each subgroup, averages were used to calculate the PUFA/MUFA ratio.

## Confocal image acquisition and analysis

Breast cancer cells were seeded onto glass coverslips ($2$–$3 \times 10^5$ per well of a 6-well plate) to have a 50% confluence. The day after manipulation cells were stained at 37 °C for (i) 15 min with BODIPY$^{493/503}$ (Thermo Fisher Scientific #D3922) to reveal LD content, or (ii) 1 h with BODIPY$^{581/591}$-C11 to evaluate lipid peroxidation (Thermo Fisher Scientific #D3861) and then fixed with 4% formaldehyde for 15 min. For nuclei staining, fixed cells were incubated with DAPI (Thermo Fisher Scientific #D3571) for 10 min at room temperature. Sample images were acquired using TCS SP8 microscope (Leica Microsystems) with LAS-AF image acquisition software. The quantification of LD was performed using CellProfiler software.

## Flow cytometry analysis

Breast cancer cells ($3$–$5 \times 10^4$ cells/well) were seeded into 12-well plates and subjected to the experimental procedure. The day after, cells were stained at 37 °C for (i) 15 min with BODIPY$^{493/503}$ to reveal LD content, or (ii) 1 h with BODIPY$^{581/591}$-C11 to evaluate lipid peroxidation. Live cells resuspended in PBS with 0.1% FBS were subjected to flow cytometry analysis using a FACSCanto II (BD Biosciences). $1 \times 10^4$ cells were analyzed for the mean fluorescence intensity (MFI) of the specific probe.

## Reactive oxygen species (ROS) analysis

Murine breast cancer cells ($3$–$5 \times 10^4$ cells/well) were seeded into 12-well plates and subjected to the experimental conditions described in Figures. Following a 24 h incubation, cells were stained with CellROX (Thermo Fisher Scientific #C10444), DCFDA (Sigma-Aldrich #D6883), or MitoSOX (Thermo Fisher Scientific #M36008) to evaluate ROS cytoplasmatic and mitochondrial content and incubated at 37 °C in the dark for 30 min. Then, cells were lysed with RIPA buffer and fluorescence was measured on a microplate reader at 485 nm excitation (Ex) and 520 nm emission (Em) for CellROX, at 485/535 nm Ex/Em for DCFDA, at 510/580 nm Ex/Em for MitoSOX.

## GSH/GSSG measurement

Murine breast cancer cells were seeded in 12-well plates ($3$–$5 \times 10^4$ cells/well) and GSH and GSSG concentration were measured after 2 h of RSL3 1 µM incubation according to manufacturer's instructions (GSH/GSSG-Glo Assay, Promega #V6611). Luminescence was read using a luminometer and was proportional to the amount of GSH. A twofold adjustment was required for GSSG concentration because each mole of oxidized GSSG upon reduction produces two moles of GSH in this assay. The GSH/GSSG ratio was calculated directly from luminescence measurements (in relative light units, RLU).

## Lipid peroxidation measurement

Malondialdehyde (MDA) concentration in 4T1 tumor-derived material was measured with the Lipid Peroxidation (MDA) Assay Kit (Sigma-Aldrich #MAK085), according to the manufacturer's instructions.

## Immunohistochemistry (IHC) analysis

4T1-derived tumors were resected, fixed in 4% paraformaldehyde (PFA), and embedded in paraffin wax. Sections of 7 µm were used and immunostained with the antibody against PLIN2 (#BS-10780R – ThermoFisher Scientific) for the IHC analysis. IHC was performed using the Leica BOND-MAX automated system (Leica Microsystems). The staining was visualized using 3,3-diaminobenzidine (DAB) and counterstained with hematoxylin. All images were captured by a slide scanner (Aperio LV1) and analyzed using ImageScope software. PLIN2 expression was evaluated using the H-score cytoplasmic assessment. Four representative fields were used for the quantification and PLIN2 expression levels. Final results were reported as cytoplasm H-score values.

## In silico analysis—analysis of human datasets

For *FADS1/2* (208964_s_at; 202218_s_at), *PLIN2* (209122_at), and *ACSL4* (202422_s_at) survival analyses, the curated dataset of ER-, PR- (assessed by immunohistochemical staining), HER2- (assessed by array) breast cancers was created using Km-plotter (http://kmplot.com) and included the relapse-free survival (RFS) data of patients belonging to the following datasets: E-MTAB-365 (Guedj et al, 2012), GSE19615 (Li et al, 2010), GSE21653 (Sabatier et al, 2011), GSE25066 (Hatzis et al, 2011), GSE2603 (Amaro et al, 2016), GSE31519 (Rody et al, 2011), GSE37946 (Liu et al, 2012), GSE45255 (Nagalla et al, 2013), GSE61304 (Grinchuk et al, 2015), GSE65194 (Maubant et al, 2015), GSE69031 (Chin et al, 2006). For *PLIN2* the curated dataset included the distant metastasis-free survival (DMFS) data of patients belonging to the following datasets: GSE19615 (Li et al, 2010), GSE25066 (Hatzis et al, 2011), GSE45255 (Nagalla et al, 2013), GSE58812 (Jézéquel et al, 2015), GSE61304 (Grinchuk et al, 2015), GSE69031 (Chin et al, 2006), GSE65194 (Maubant et al, 2015). For *FADS1/2* the curated dataset included the overall survival (OS) data of patients belonging to the following datasets: GSE81538 (Brueffer et al, 2018), GSE96058 (Brueffer et al, 2018). For *FADS1/2* survival analysis, the weighted (1:1) mean expression of the selected genes was used and patients were dichotomized into high and low expressing using the best cutoff threshold (Lánczky and Győrffy, 2021).

## Statistics and reproducibility

Statistics were performed using Prism 10 (GraphPad Software). Lipidomics statistical analysis, heatmaps, and volcano plots were made with Metaboanalyst 5.0 (https://www.metaboanalyst.ca/). Unless stated otherwise, all numerical data are expressed as the mean ± standard error of the mean (SEM). All experiments were conducted at least 3 times independently, with one or more technical replicates for each experimental condition tested. Unless stated otherwise, comparisons between 2 groups were made using the two-tailed, unpaired Student's t-test. Comparisons between multiple groups were made using one-way or two-way analysis of variance (ANOVA). Bonferroni, Dunnett, Tukey, or Šidák post-testing analyses with a confidence interval of 95% were used for individual comparisons as reported in figure legends. Multivariate Cox analyses on the cohort of patients analyzed were generated using KM-plotter. Statistical significance was defined when

**The paper explained**

**Problem**

Triple-negative breast cancer (TNBC) is characterized by rapid progression and metastases and poses a significant clinical challenge as it lacks prognostic and predictive markers together with effective therapies, necessitating a deeper understanding of its biology to identify potential targets for intervention.

**Results**

Our studies reveal a dysregulated lipid metabolism in TNBC characterized by aggressive traits. We also identify the prognostic significance of desaturases FADS1 and FADS2 in this context. Importantly, the higher expression of FADS1 and FADS2 distinguished a subset of TNBC vulnerable to ferroptosis induction.

**Impact**

Our findings describe the complex metabolic reprogramming underlying the aggressive nature of TNBC and offer stratification biomarkers to improve the clinical management of TNBC patients. In addition, we present evidence supporting targeting strategies that involve dysregulation of lipid metabolism and induction of ferroptosis in preclinical models, which warrant further exploration in clinical trials.

$P < 0.05$, with the exact $p$-value reported when $P > 0.0001$ or indicated as $P < 0.0001$ for any $p$-value below that threshold. $P$-values are reported only when biologically relevant, as indicated in Figure Legends. When differences were not statistically significant or the comparison not biologically relevant, no indication was reported in the figures. The treatment groups were randomized, but blinding was not implemented during the experimental processes. No sample/animal was excluded from the study.

## Data availability

The transcriptomic data uploaded to GEO database can be accessed at GSE236033.

The source data of this paper are collected in the following database record: biostudies:S-SCDT-10_1038-S44321-024-00090-6.

## Peer review information

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

## Acknowledgements

The work was funded by *Fondazione Associazione Italiana Ricerca sul Cancro* (AIRC) *ETS* and *Fondazione Cassa di Risparmio di Firenze* (grant Multiuser 19515 to PC and AM, grant IG 22941 to AM, grant IG 8797 to PC), PNRR M4C2-Investimento 1.4-CN00000041 CN3 "*Sviluppo di Terapia Genica e Farmaci con Tecnologia ad RNA*" funded by NextGenerationEU (to EG, and AM), PNRR "THE - Tuscany Health Ecosystem" *ambito di intervento* "1. Health" ECS00000017 funded by NextGenerationEU (to MB, PC, and AM). NL was supported by an AIRC fellowship. AS is supported by *Associazione Annastaccatolisa*. LI is supported by *Fondazione Pezcoller*. We extend our thanks to *Fondazione Radioterapia Oncologica* (FRO) nonprofit organization and all those who have provided essential support for scientific research. We thank Prof Clare Isacke (The Institute of Cancer Research, London) for the fruitful discussion and insightful comments. The data presented in the current study were in part generated using the equipment of the *Facility di Medicina Molecolare*, funded by "*Ministero dell'Istruzione dell'Università e della Ricerca – Bando Dipartimenti di Eccellenza 2018-2022*". The illustration was created with BioRender.com.

## Author contributions

**Nicla Lorito**: Conceptualization; Data curation; Formal analysis; Validation; Investigation; Methodology; Writing—review and editing. **Angela Subbiani**: Data curation; Formal analysis; Investigation; Methodology; Writing—review and editing. **Alfredo Smiriglia**: Investigation; Methodology; Writing—review and editing. **Marina Bacci**: Formal analysis; Investigation; Writing—review and editing. **Francesca Bonechi**: Investigation; Writing—review and editing. **Laura Tronci**: Investigation; Methodology; Writing—review and editing. **Elisabetta Romano**: Investigation; Methodology. **Alessia Corrado**: Investigation; Methodology. **Dario Livio Longo**: Investigation; Methodology. **Marta Iozzo**: Investigation; Methodology. **Luigi Ippolito**: Methodology; Writing—review and editing. **Giuseppina Comito**: Investigation; Methodology. **Elisa Giannoni**: Formal analysis; Methodology; Writing—review and editing. **Icro Meattini**: Formal analysis; Methodology; Writing—review and editing. **Alexandra Avgustinova**: Data curation; Investigation; Writing—review and editing. **Paola Chiarugi**: Formal analysis; Methodology; Writing—review and editing. **Angela Bachi**: Investigation; Methodology; Writing—review and editing. **Andrea Morandi**: Conceptualization; Formal analysis; Supervision; Funding acquisition; Investigation; Methodology; Writing—original draft; Project administration; Writing—review and editing.

Source data underlying figure panels in this paper may have individual authorship assigned. Where available, figure panel/source data authorship is listed in the following database record: biostudies:S-SCDT-10_1038-S44321-024-00090-6.

## Disclosure and competing interests statement

IM declares consultant honoraria for Eli Lilly, Novartis, Seagen, Istituto Gentili, Roche, Pfizer, Ipsen, and Pierre Fabre. All the other authors declare no competing interests.

# Expanded View Figures

**Figure EV1. Details of altered lipid metabolism in aggressive TNBC cells, related to Fig. 1.** ▶

(A) 67NR, 4T07, and 4T1 breast cancer cells were cultured ON in a medium containing $^{14}$C-U-(uniformly) radioactively labeled glucose, lactate, acetate, or glutamine. Lipids were extracted and the radioactive signal was measured to monitor the amount of each metabolite that is incorporated into lipids, as described in the Materials and Methods section. Each value was normalized on protein content ($n = 3$ biological replicates in either single, duplicate, triplicate or more than three technical replicates). The 67NR cell line was used as comparator in the statistical analysis. (B, C) Murine 4T1 (B) and D2A1 (C) breast cancer cell series were analyzed by quantitative real-time polymerase chain reaction (qRT-PCR) analysis using the assays described in the figure. Fold relative enrichment is shown using the non-metastatic cells as comparator ($n = 3$ biological replicates in either single or technical duplicate). The 67NR or the D2A1 cells were used as comparator in the statistical analysis. (D) The 4T1, D2A1, and human TNBC cells were subjected to cytofluorimetric analysis. FACS analysis of the mean fluorescence intensity (MFI) of the populations positive for BODIPY$^{493/503}$ was reported ($n = 3$ biological replicates in either single or technical triplicate). The non-metastatic cells were used as comparators in the statistical analysis. (E) Murine 4T1 and D2A1 series were analyzed by qRT-PCR analysis using the assay described in the figures. Fold relative enrichment is shown using the 67NR or the D2A1 cells as comparator ($n = 3$ biological replicates in either single or technical duplicate). The 67NR or the D2A1 cells were used as comparator in the statistical analysis. (F, G) Seahorse XFe96 Mito Stress Test was performed on 4T1 cell line series treated with 2 μM UK5099 (mitochondrial pyruvate carrier, MPC, inhibitor), 3 μM BPTES (glutaminase 1, GLS-1, inhibitor), or 4 μM Etomoxir (CPT1A inhibitor) for 30 min in the presence of standard condition (full medium), and oxygen consumption rate (OCR) was calculated in real-time after the administration of the ATP synthase inhibitor oligomycin, the proton uncoupler carbonyl cyanide p-triflouromethoxyphenylhydrazone (FCCP), and the respiratory complex I inhibitor rotenone together with the respiratory complex III inhibitor antimycin A (Rot/AA) (F). Basal and maximal respiration was calculated as described in the Materials and Methods section and normalized on protein content (G) ($n = 3$ biological replicates in either single, duplicate, triplicate, or more than three technical replicates). The untreated (NT) condition was used as comparator in the statistical analysis. (H) Murine TNBC cells were treated with 40 μM of Etomoxir for 30 min. After detachment, cells were subjected to high-resolution respirometry analysis by the Oroboros-O2K instrument. Left: Representative graphs of cell respirometry analysis in the control (up) and treatment (down) conditions. The blue curve represents the oxygen concentration, whereas the red slope shows the oxygen consumption before and after the serial injections of oligomycin (O), uncoupler CCCP (C), and Antimycin A (A). Right: Bar chart graph of basal oxygen consumption (Routine), proton leak (Leak), and maximal oxygen consumption (E) values subtracted from residual oxygen consumption (ROX) in control and Etomoxir treated cells ($n = 3$ biological replicates). Data information: data are presented as mean ± SEM. Statistical analysis was performed using one-way ANOVA followed by Tukey's correction (A, G) or Dunnett's correction (B–E), two-way ANOVA followed Bonferroni's correction (F) or Tukey's correction (H). Source data are available online for this figure.

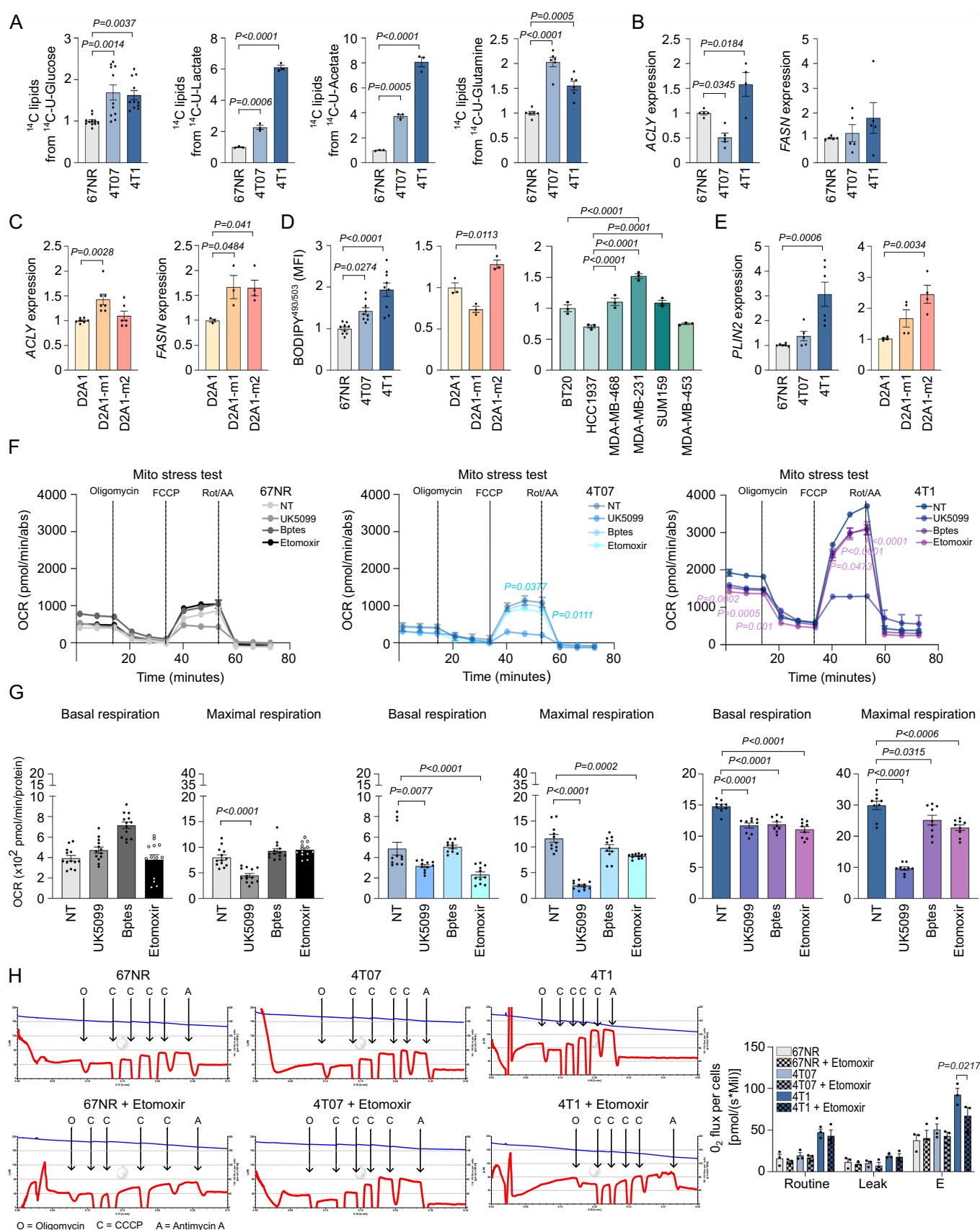

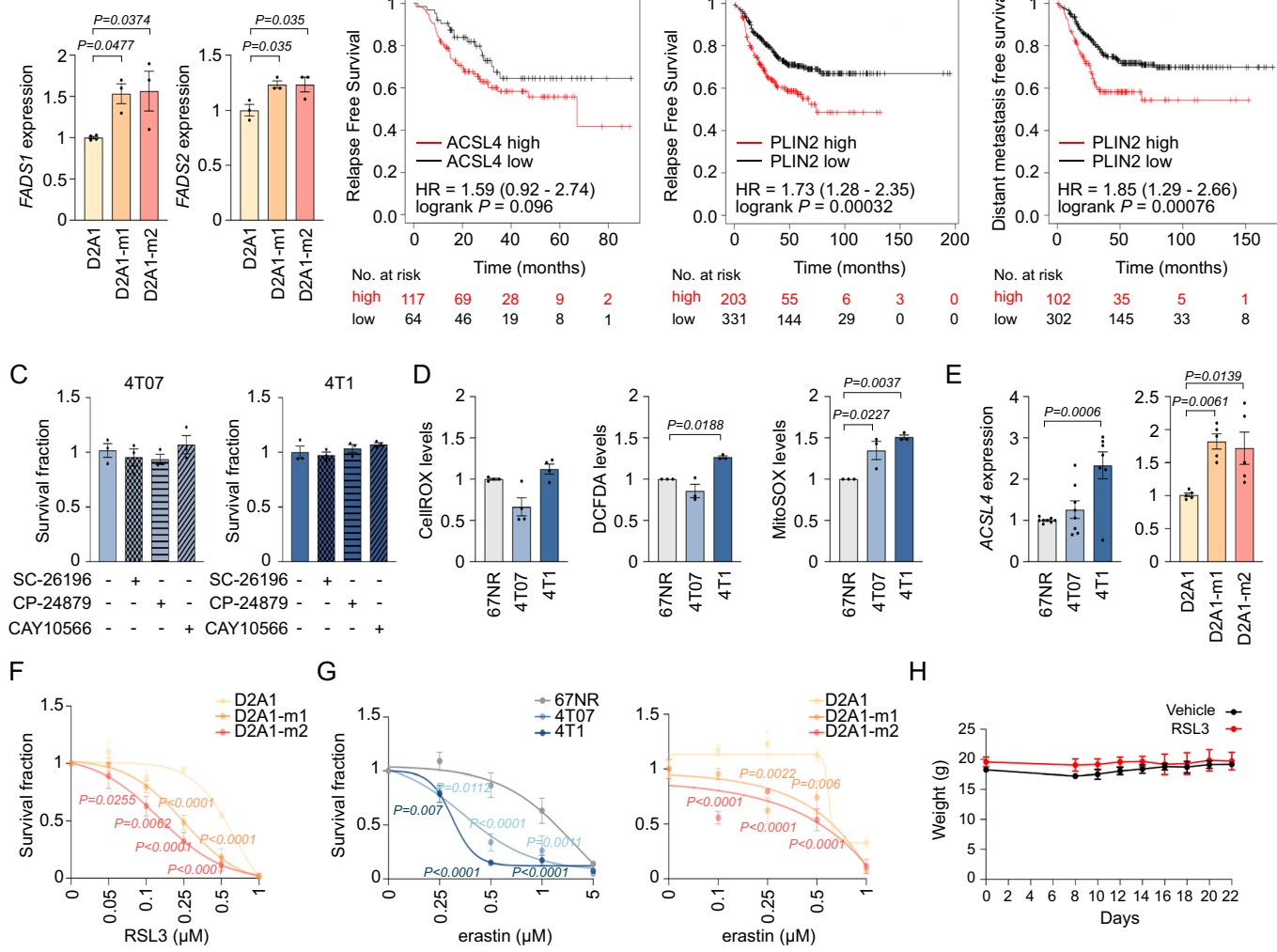

**Figure EV2.  Details of FADS1 and FADS2 expression in aggressive TNBC and their susceptibility to ferroptosis induction, related to Fig. 2.**

(**A**) D2A1 cell line series were analyzed by qRT-PCR analysis using the assays described in the figure. Fold relative enrichment is shown using the D2A1 cells as comparator ($n = 3$ biological replicates in either single or technical duplicate). The D2A1 cells were used as comparator in the statistical analysis. (**B**) Kaplan–Meier analysis of RFS and DMFS of a curated cohort of TNBC patients divided into high and low for *ACSL4* or *PLIN2* expression as described in the Materials and Methods section. HR and log-rank Mantel-Cox *P* values are shown. (**C**) TNBC metastatic 4T07 and 4T1 cells were treated with 10 μM FADS2i (SC-26196), FADS1/2i (CP-24879), or SCD1i (CAY10566) for 24 h and subjected to cell viability assay ($n = 3$ biological replicates). (**D**) Intracellular ROS levels were measured by CellROX and DCFDA staining while mitochondrial ROS levels by MitoSOX in TNBC 4T1 series' cells ($n = 3$ biological replicates in either single or technical duplicate). The 67NR cell line was used as comparator in the statistical analysis. (**E**) Murine TNBC cells were analyzed by qRT-PCR analysis using the assay described in the figure ($n = 3$ biological replicates in either single, duplicate, or technical triplicate). Fold relative enrichment and statistical analysis are shown using the non-metastatic (67NR or D2A1) cells as comparator. (**F, G**) 24-h dose-response curve of RSL3 (**F**) and erastin (**G**) showed a differential effect between less aggressive and metastatic cells of the D2A1 or 4T1 series ($n = 3$ biological replicates). Statistics is shown using the non-metastatic (67NR or D2A1) cells as comparator. (**H**) Weight of BALB/c mice exposed for 15 days to 40 mg/kg RSL3 ($n = 4$–6 mice/ group). Data information: data are presented as mean ± SEM. Statistical analysis was performed using one-way ANOVA followed by Dunnett's correction (**A, C, E**) or Tukey's correction (**D**), two-way ANOVA followed by Bonferroni's correction (**F–H**). Source data are available online for this figure.

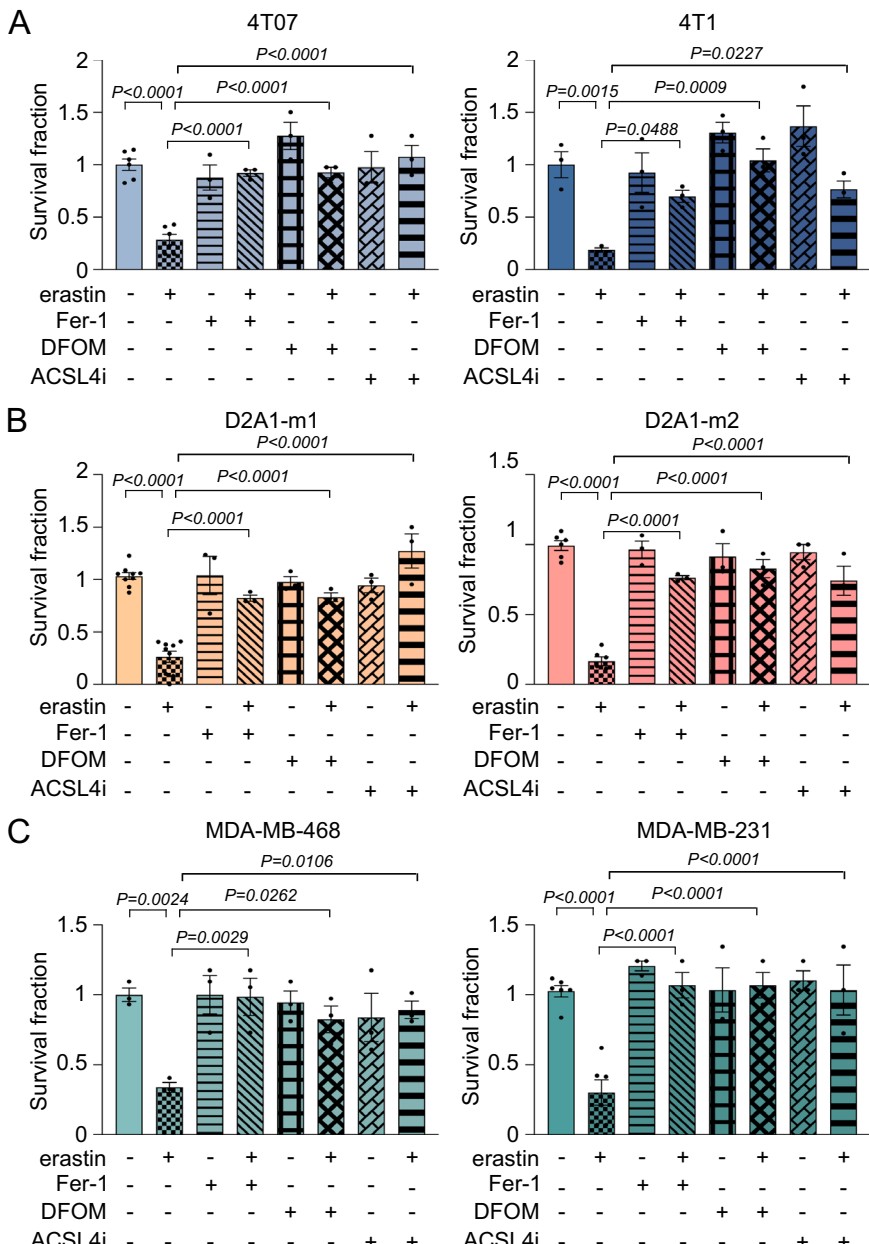

**Figure EV3. Details of Ferrostatin-1, Deferoxamine, and ACSL4 inhibition ability to prevent RSL3-induced cell death in TNBC cells, related to Fig. 3.**

(A–C) Metastatic 4T07 and 4T1 (A), D2A1-m1 and D2A1-m2 (B), MDA-MB-468 and MDA-MB-231 (C) cells were pre-treated with 15 μM Fer-1, 5 μM DFOM, or 10 μM ACSL4i for 4 h and then exposed ON to 0.5 μM erastin. After 24 h cells were subjected to cell viability assay (n = 3 biological replicates in either single, duplicate, or technical triplicate). The erastin-treated condition was used as comparator in the statistical analysis. Data information: data are presented as mean ± SEM. Statistical analysis was performed using one-way ANOVA, Dunnett corrected. Source data are available online for this figure.

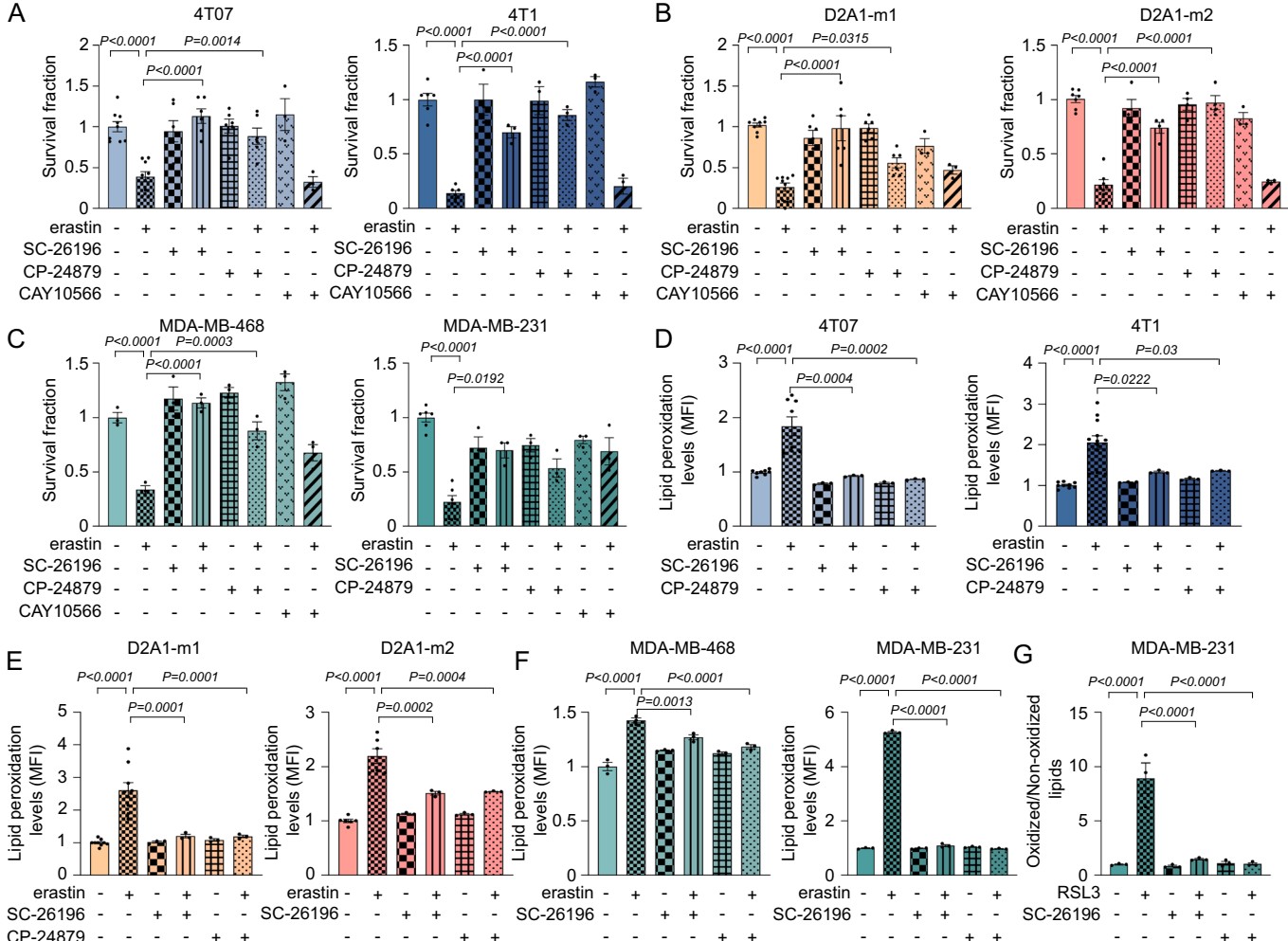

**Figure EV4. Details of the ability of FADS1/2 targeting in preventing ferroptosis induction in aggressive TNBC, related to Fig. 4.**

(A–C) TNBC metastatic 4T07 and 4T1 (A), D2A1-m1 and D2A1-m2 (B), human MDA-MB-468 and MDA-MB-231 (C) cells were pre-treated with 10 μM FADS2i (SC-26196), FADS1/2i (CP-24879), and SCD1i (CAY10566) for 4 h, exposed ON to 0.5 μM erastin, and subjected to cell viability assay (n = 3 biological replicates in either single, duplicate or technical triplicate). The erastin-treated condition was used as comparator in the statistical analysis. (D–F) TNBC metastatic 4T07 and 4T1 (D), D2A1-m1 and D2A1-m2 (E), human MDA-MB-468 and MDA-MB-231 (F) cells were pre-treated with 10 μM FADS2i (SC-26196), FADS1/2i (CP-24879), and SCD1i (CAY10566) for 4 h, exposed for 2 h to 5 μM erastin, and subjected to cytofluorimetric analysis to measure lipid peroxidation (n = 3 biological replicates in either single, duplicate or technical triplicate). The erastin-treated condition was used as comparator in the statistical analysis. (G) Highly metastatic MDA-MB-231 cells were pre-treated with 10 μM FADS2i or FADS1/2i for 4 h, exposed for 2 h to 1 μM RSL3, and subjected to confocal analysis to measure lipid peroxidation. Relative quantification is shown (n = 3 biological replicates). The RSL3-treated condition was used as comparator in the statistical analysis. Data information: data are presented as mean ± SEM. Statistical analysis was performed using one-way ANOVA followed by Dunnett's correction (A–F) or Tukey's correction (G). Source data are available online for this figure.

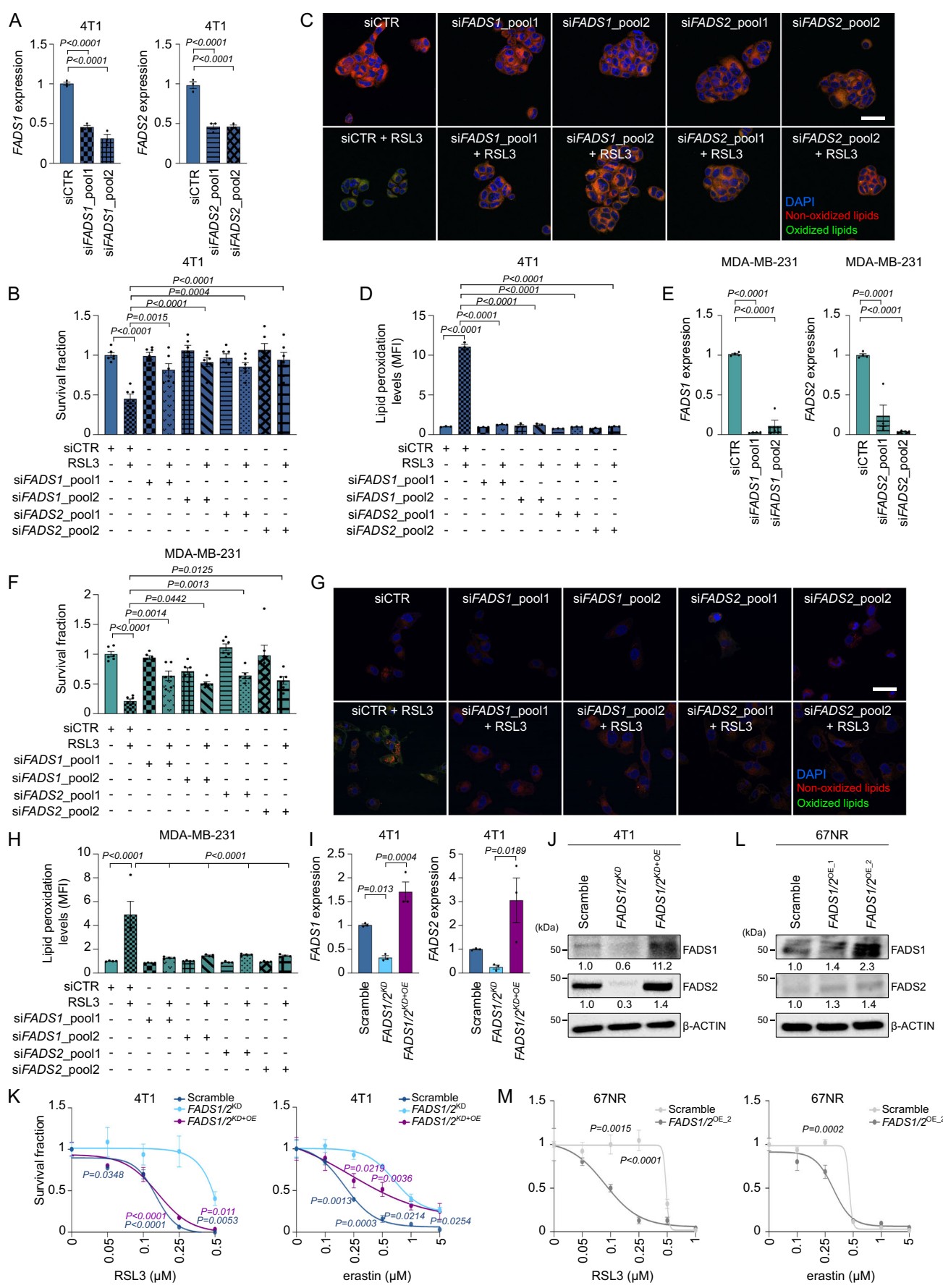

◀

**Figure EV5.    Details of FADS1/2 knockdown effect on lipidomic profile of TNBC cells and their susceptibility to ferroptosis induction, related to Fig. 5.**

(A–D) 4T1 cells transfected with non-targeting small interfering RNA (siCTR) or 2 different combination pools of 4 individual siRNA for FADS1 (siFADS1_pool1 and pool2) and FADS2 (siFADS2_pool1 and pool2) and assessed by qRT-PCR (**A**) were treated ON with 0.1 µM RSL3 (**B**) or 2 h with 1 µM RSL3 (**C, D**) and assayed for cell survival (**B**), confocal analysis (**C**), and cytofluorimetric analysis (**D**) to measure lipid peroxidation. Representative pictures of BODIPY[581/591]-C11 stained cells are shown (oxidized lipids: green; non-oxidized lipids: red; nuclei: blue, DAPI; scale bar, 10 µm) ($n = 3$ biological replicates in either single or technical duplicate). siCTR transfected cells (**A**) and siCTR transfected cells treated with RSL3 (**B, D**) were used as comparator in the statistical analysis. (E–H) MDA-MB-231 cells transfected with siCTR or 2 different combination pools of 4 individual siRNA for FADS1 (siFADS1_pool1 and pool2) and FADS2 (siFADS2_pool1 and pool2) and assessed by qRT-PCR (**E**) were treated ON with 0.25 µM RSL3 (**F**) or 2 h with 1 µM RSL3 (**G, H**) and assayed for cell survival (**F**), confocal analysis (**G**), and cytofluorimetric analysis (**H**) to measure lipid peroxidation. Representative pictures of BODIPY[581/591]-C11 stained cells are shown (oxidized lipids: green; non-oxidized lipids: red; nuclei: blue, DAPI; scale bar, 10 µm) ($n = 3$ biological replicates in either single or technical duplicate). siCTR transfected cells (**E**) and siCTR transfected cells treated with RSL3 (**F, H**) were used as comparator in the statistical analysis. (I–K) Stable *FADS1/FADS2* double knockdown transfected 4T1 cells (FADS1/2[KD] 4T1) and the corresponding cells with the ectopic re-introduction of FADS1 and FADS2 (FADS1/2[KD+OE] 4T1), whose FADS1/2 expression was assessed using qRT-PCR and WB analyses (**I, J**), were exposed to increasing concentrations of RSL3 or erastin in a dose-response curve assay (**K**) ($n = 3$ biological replicates). The FADS1/2[KD] condition was used as comparator in the statistical analysis. (L, M) 67NR cells over-expressing both FADS1 and FADS2, FADS1/2[OE], (**L**) were grown for 24 h in the presence of increasing doses of RSL3 and erastin (**M**) before assaying cell viability ($n = 3$ biological replicates). Data information: In (**A, B, D–F, H, I, K, M**) data are presented as mean ± SEM. Statistical analysis was performed using one-way ANOVA followed by Dunnett's correction (**A, B, D–F, H, I**) or two-way ANOVA followed by Bonferroni's correction (**K, M**). Source data are available online for this figure.

