## [Peer Review File · EMBO Molecular Medicine]

FADS1/2 control lipid metabolism and ferroptosis susceptibility in triple-negative breast cancer

Nicla Lorito, Angela Subbiani, Alfredo Smiriglia, Marina Bacci, Francesca Bonechi, Laura Tronci, Elisabetta Romano, Alessia Corrado, Dario Livio Longo, Marta Iozzo, Luigi Ippolito, Giuseppina Comito, Elisa Giannoni, Icro Meattini, Alexandra Avgustinova, Paola Chiarugi, Angela Bachi, and Andrea Morandi

Corresponding author(s): Andrea Morandi (andrea.morandi@unifi.it)

Review Timeline:

Submission Date:	30th Jun 23
Editorial Decision:	13th Jul 23
Revision Received:	17th Apr 24
Editorial Decision:	6th May 24
Revision Received:	24th May 24
Accepted:	31st May 24

Editor: Lise Roth

Transaction Report:

13th Jul 2023

Dear Prof. Morandi,

Thank you for the submission of your manuscript to EMBO Molecular Medicine. We have now received feedback from the three reviewers who agreed to evaluate your manuscript. As you will see from the reports below, the referees acknowledge the interest of the study and are overall supporting publication of your work pending appropriate revisions.

Upon further discussion, the referees agreed that additional in vivo work would be welcome but not necessary for acceptance of the manuscript. However, expansion to additional in vitro models will be required. Similarly, human relevance will have to be clearly shown in the revised version of your manuscript.

As revising the manuscript according to the referees' recommendations appears to require a lot of additional work and experimentation, and given the potential interest of your findings, we are ready to extend the deadline to 6 months with the understanding that acceptance of the manuscript would entail a second round of review.

EMBO Molecular Medicine encourages a single round of revision only and therefore, acceptance or rejection of the manuscript will depend on the completeness of your responses included in the next, final version of the manuscript. For this reason, and to save you from any frustrations in the end, I would strongly advise against returning an incomplete revision.

We require:

4) A .docx formatted letter INCLUDING the reviewers' reports and your detailed point-by-point responses to their comments. As part of the EMBO Press transparent editorial process, the point-by-point response is part of the Review Process File (RPF), which will be published alongside your paper.

5) A complete author checklist, which you can download from our author guidelines (<https://www.embopress.org/page/journal/17574684/authorguide#submissionofrevisions>). Please insert information in the checklist that is also reflected in the manuscript. The completed author checklist will also be part of the RPF.

6) Please note that all corresponding authors are required to supply an ORCID ID for their name upon submission of a revised manuscript.

7) It is mandatory to include a 'Data Availability' section after the Materials and Methods. Before submitting your revision, primary datasets produced in this study need to be deposited in an appropriate public database, and the accession numbers and database listed under 'Data Availability'. Please remember to provide a reviewer password if the datasets are not yet public (see <https://www.embopress.org/page/journal/17574684/authorguide#dataavailability>).

8) For data quantification: please specify the name of the statistical test used to generate error bars and P values, the number (n) of independent experiments (specify technical or biological replicates) underlying each data point and the test used to calculate p-values in each figure legend. The figure legends should contain a basic description of n, P and the test applied. Graphs must include a description of the bars and the error bars (s.d., s.e.m.). Please provide exact p values.

9) Our journal encourages inclusion of *data citations in the reference list* to directly cite datasets that were re-used and obtained from public databases. Data citations in the article text are distinct from normal bibliographical citations and should

directly link to the database records from which the data can be accessed. In the main text, data citations are formatted as follows: "Data ref: Smith et al, 2001" or "Data ref: NCBI Sequence Read Archive PRJNA342805, 2017". In the Reference list, data citations must be labeled with "[DATASET]". A data reference must provide the database name, accession number/identifiers and a resolvable link to the landing page from which the data can be accessed at the end of the reference. Further instructions are available at .

13) Author contributions: CRediT has replaced the traditional author contributions section because it offers a systematic machine readable author contributions format that allows for more effective research assessment. Please remove the Authors Contributions from the manuscript and use the free text boxes beneath each contributing author's name in our system to add specific details on the author's contribution. More information is available in our guide to authors.

16) As part of the EMBO Publications transparent editorial process initiative (see our Editorial at <http://embomolmed.embopress.org/content/2/9/329>), EMBO Molecular Medicine will publish online a Review Process File (RPF) to accompany accepted manuscripts.

In the event of acceptance, this file will be published in conjunction with your paper and will include the anonymous referee reports, your point-by-point response and all pertinent correspondence relating to the manuscript. Let us know whether you agree with the publication of the RPF and as here, if you want to remove or not any figures from it prior to publication.

I look forward to receiving your revised manuscript.

Yours sincerely,

Lise Roth

***** Reviewer's comments *****

Referee #1 (Comments on Novelty/Model System for Author):

As noted in the review below, the experiments need to be repeated in human cell lines to ensure they are relevant, and more detailed analysis of human patient data needs to be included. siRNA data use only a single siRNA, 2 should be used to limit potential off target effects.

Referee #1 (Remarks for Author):

Overall this paper is interesting in light of recent data on the role of ferroptosis in TNBC. The authors add to the current literature in determining that FADS1/2 are important in this process, however there are a number of additional experiments and analyses that would be required prior to publication. In particular, the authors should undertake more analysis using human cell lines, and utilise the recent literature and datasets to better incorporate their study of mouse cell lines with human data and human clinical samples.

There is a disparity between the 4T1 and 4T07 data in Figure 1, whereby 14C-glucose contributes to an increase in lipids in both cell lines over the 67NR, however there are significant differences in lipid droplet accumulation as well as associated gene expression for lipid synthesis between 4T1 and 4T07. Is this due to generation of lipids from sources other than glucose in 4T1, or differences in lipid storage in the 4T1 cells? This should be examined in more detail.

The D2A1 data are interesting, showing again differences in the two metastatic lines in terms of ACLY expression. It would be helpful to see the same lipid droplet data and PLIN2 expression for these three cell lines to see how they compare together, and with the 4T1 data.

These are all mouse models. Analysis of a panel of TNBC cell lines (including LAR cell lines) as detailed would be helpful in undertaking some of the key experiments performed in the mouse cell lines throughout this study. For these data to be relevant to human disease, as suggested by the KMplot data, this is essential. While it is appreciated that this is done in part in Figure S3g, h, i, j, k, additional cell lines and assays should be included here, rather than just a single line MDA-MB-231 for most assays. At a minimum this should include analysis and comparison of ACSL4, FADS1 and FADS2 levels by western blot, along with SCD1, SCD2, ACLY, FASN, PLIN2. Induction of ferroptosis with RSL3 and Erastin, and inhibition of key pathways using Fer-1, SC26196 and CP24879 at a minimum. These data should include analysis using BODIPY assays to assess lipid droplets and peroxidation as per the mouse experiments. It would also be important to undertake knockdown or knockout for FADS1 AND FADS2 in human cell lines, and to use more than a single siRNA as currently done in the mouse cells. 2 separate targeting siRNAs should be used.

It is somewhat frustrating that many of the assays throughout this paper do not include the control cell lines (67NR or BT20) for key assays, where having a non-responsive control would be helpful to show the specificity of the inhibitors for the highly metastatic/aggressive cells.

The in vivo experiment in Figure 2L should have had analyses of the tumours using a variety of assays such as looking for lipid droplets, lipidomics etc to assess the mechanism by with RSL3 decreased tumour growth. Comparison with 67NR cells in some of the in vitro (Figure 2J/k assays) as well as this in vivo assay with RSL3 would also have been helpful to show that this is specific to the more aggressive lines.

Care should be taken throughout to not overstate the data for 4T07 and 4T1.

4T07 had no significant change in basal respiration on Seahorse or using the Oroboros data, and the text should reflect that these changes were only consistent in the 4T1 cells.

It is unclear how Supp Fig S1c-e show "enhanced oxidative metabolism" for 4T07, as their basal OCR does not appear different to 67NR (better quantified in f-h).

There is no significant potentiation in 4T07 cells with either inhibitor as suggested: "Although 4T07 and 4T1 cells were already sensitive to RSL3, impairing LD biogenesis using the DGAT1 and DGAT2 inhibitors potentiated the cell death induced by RSL3, particularly in the more aggressive 4T1 cells (Figure 3d)."

CAY10566 inhibition is missing from Fig S3A and Erastin is should be included in Fig S3B/C analyses.

The authors should access the lipidomics data published as part of the Xiao et al Cell Research 32, 477-490 (2022) manuscript. They should compare their data in Figure 4 with these data to determine whether there is a lipid signature similar to human patient data.

The in silico analysis should be extended to examine the various TNBC subsets as per the Yang et al 2023 Cell Metabolism paper. Are FADS1 and FADS2 higher in the LAR subset? What about the other genes examined? Correlation analyses could also be performed for each of the key genes examined in this study to determine whether they are co-regulated.

Referee #2 (Comments on Novelty/Model System for Author):

This study builds upon previous research, that has demonstrated the heightened sensitivity of triple-negative breast cancers (TNBCs) to ferroptosis. While this aspect has been established for several years, it alone may not provide the desired novelty. The intriguing connection between ferroptosis sensitivity and FADS1/2 deserves attention; however, the data presented in this study is perplexing and primarily relies on siRNA-based approaches, which do not assess the transient effect or its sufficiency in inducing the expected lipidomic changes. Therefore, the inclusion of knockout models is crucial to establish firm conclusions.

Referee #2 (Remarks for Author):

This paper elucidates a mechanism that renders aggressive triple-negative breast cancer cells susceptible to ferroptosis, a form of cell death triggered by lipid peroxidation. The study highlights the significance of FADS1/2 desaturases in regulating intracellular levels of polyunsaturated fatty acids (PUFAs), which are crucial for lipid peroxidation and found in elevated levels in aggressive triple-negative breast cancers. The findings propose that FADS1/2-mediated lipid metabolic reprogramming drives ferroptosis sensitivity in triple-negative breast cancer. While the study offers valuable insights, addressing the following concerns would provide stronger support for the data and claims presented.

Validation of RSL3 as a GPX4 Inhibitor in vivo:

Given the known instability of RSL3 in vivo, the utilization of RSL3 as a GPX4 inhibitor in in vivo experiments needs reconciliation. The authors should provide validation that RSL3 can effectively act as a GPX4 inhibitor in vivo, potentially through liproxstatin rescue as a viable alternative.

Unquestionable Proof of FADS1/2 Function:

As the central focus of this work, the presented data concerning FADS1/2 requires further substantiation. To establish an incontrovertible role for FADS1/2, the authors should employ FADS1/FADS2 double deficient cells along with corresponding add-back controls. It is crucial to test the titration of RSL3/Erastin over a range of doses rather than a single point to obtain comprehensive evidence. Furthermore, utilizing such models would allow the authors to validate the on-target effects of the inhibitors used in the study. While not mandatory for this study, establishing the role of FADS1/2 in in vivo models would provide additional support.

Characterization of Lipidome in FADS-Deficient Cells:

To reinforce the proposed role of FADS1/2, it is imperative to characterize the lipidome of FADS-deficient cells in the cell lines where the authors claim its involvement. This analysis would provide valuable insights into the lipid composition changes associated with the alleged role of FADS1/2 in the studied cell lines.

Validation of FADS1/2 Function in Human Breast Cancer Cell Lines:

Strengthening the conclusions can be achieved by validating the role of FADS1/2 in a few human breast cancer cell lines. By demonstrating consistency across multiple cell lines, the findings would gain broader applicability and enhance the overall robustness of the study.

Exploring Potential Interactions with ER Antagonists:

A recent report has proposed that ER antagonists sensitize ER+ breast cancer to ferroptosis by downregulating MBOAT1 (PMID 37267948). Considering the straightforward use of ER antagonists, the authors should explore whether such a mechanism impacts the FADS1/2 axis as well. Investigating this potential interaction would provide additional insights and enrich the study's scope.

In-depth Discussion on the Advantages of Increased PUFA in TNBC Growth:

While the authors briefly discuss the potential advantages of increased PUFA levels in triple-negative breast cancer growth, a more comprehensive and thoughtful discussion on this aspect is warranted. Such a discussion would promote further discourse on the reasons why cells with heightened sensitivity to ferroptosis thrive, adding depth to the findings and stimulating additional research avenues.

Addressing these concerns would strengthen the authors' data and claims, augmenting our understanding of triple-negative breast cancer and ferroptosis.

Referee #3 (Comments on Novelty/Model System for Author):

In the manuscript, the authors identify FADS1/2 is highly expressed in TNBC. after the related analysis (lipidomic, metabolic, and other assays), The authors found that FADS1/2 high-expressing TNBC are susceptible to ferroptosis. Combination of targeting FADS1/2 and ferroptosis-inducing agents may provide new therapy for TNBC patients. This study provided a new therapy for TNBC patients. However, there are too many issues which need to be addressed.

1. The whole manuscript is difficult to follow. So, the authors should make it smooth and easy to be understood.
2. the data quality needs to be improved (e.g., fig.1d). There are too many data which is not so convincing or consistent.
3. For the cell models (Fig 1a), the authors should provide the data to support their aggressive ability.
4. the authors should provide a figure to summary their finding in Fig.6.
5. The manuscript should enhance the novelty.
- 4.

Referee #3 (Remarks for Author):

suited to publication as a short report in EMBO Molecular Medicine, but need to be improved.

***** Reviewer's comments *****

Referee #1

(Comments on Novelty/Model System for Author):

As noted in the review below, the experiments need to be repeated in human cell lines to ensure they are relevant, and more detailed analysis of human patient data needs to be included. siRNA data use only a single siRNA, 2 should be used to limit potential off-target effects.

The majority of the experiments described in the study have been now reproduced in 6 human cell lines and the siRNA approach has been adjusted by using individual siRNA pooled in 2 different combinations, i.e., we have used in total 4 different siRNA for each gene and reported the 2 siRNA pools in the new experiments that have been described in the revised form of the manuscript. Finally, an analysis of pertinent human patient datasets, including a triple-negative breast cancer repository previously stratified into distinct TNBC subcategories, has been conducted. These recent data of clinical significance have been incorporated into the updated version of the manuscript (refer to subsequent sections for comprehensive elucidation).

(Remarks for Author):

Overall this paper is interesting in light of recent data on the role of ferroptosis in TNBC. The authors add to the current literature in determining that FADS1/2 are important in this process, however there are a number of additional experiments and analyses that would be required prior to publication. In particular, the authors should undertake more analysis using human cell lines, and utilise the recent literature and datasets to better incorporate their study of mouse cell lines with human data and human clinical samples.

A large part of the study has been reproduced in 6 human cell lines and the analysis of human patient data has been improved and incorporated in the study (**Figures 1-4 and EV1-EV5, Appendix S1-S3**).

There is a disparity between the 4T1 and 4T07 data in Figure 1, whereby ¹⁴C-glucose contributes to an increase in lipids in both cell lines over the 67NR, however there are significant differences in lipid droplet accumulation as well as associated gene expression for lipid synthesis between 4T1 and 4T07. Is this due to generation of lipids from sources other than glucose in 4T1, or differences in lipid storage in the 4T1 cells? This should be examined in more detail.

This is an interesting observation. We have analyzed alternative potential sources by using ¹⁴C-labeled lactate, acetate, and glutamine and found that 4T1 and 4T07 have indeed a different use of these nutrients (i.e., 4T1 cells have a superior capacity to fuel lipogenesis through lactate and acetate that may explain their greater accumulation of lipid droplets). These new data have been included in the new version of the manuscript (**Figure EV1A**).

The D2A1 data are interesting, showing again differences in the two metastatic lines in terms of ACLY expression. It would be helpful to see the same lipid droplet data and PLIN2 expression for these three cell lines to see how they compare together, and with the 4T1 data.

We thank the Reviewer for the suggestion. We have fully characterized the D2A1 series and measured lipid droplet content, PLIN2, ACLY, and FASN protein and mRNA expression levels. These new data have been included in the revised version of the manuscript (**Figure 1C-E and Figure EV1C-E**).

These are all mouse models. Analysis of a panel of TNBC cell lines (including LAR cell lines) as detailed would be helpful in undertaking some of the key experiments performed in the mouse cell lines throughout this study. For these data to be relevant to human disease, as suggested by the KMplot data, this is essential. While it is appreciated that this is done in part in Figure S3g, h, i, j, k, additional cell lines and assays should be included here, rather than just a single line MDA-MB-231 for most assays. At a minimum this should include analysis and comparison of ACSL4, FADS1 and FADS2 levels by western blot, along with SCD1, SCD2, ACLY, FASN, PLIN2. Induction of ferroptosis with RSL3 and Erastin, and inhibition of key pathways using Fer-1, SC26196 and CP24879 at a minimum. These data should include analysis using BODIPY assays to assess lipid droplets and peroxidation as per the mouse experiments. It would also be important to undertake knockdown or knockout for FADS1 AND FADS2 in human cell lines, and to use more than a single siRNA as currently done in the mouse cells. 2 separate targeting siRNAs should be used.

This has been a challenging request, but we agree with the Reviewer that including relevant human models has strengthened the message and the implication of the findings described in the study. We have indeed repeated all the assays requested by the Reviewer (at a minimum) and extended the analysis with a particular focus on those functionally essential experiments in a panel of 6 additional human cell lines, including a LAR cell line (as asked), being particularly conscious in selecting the different types of TNBC cells that have been described to have different tropism for secondary sites, i.e., metastatic potential (Jin et al., Nature 2020, PMID: 33299191). The siRNA approach described above (i.e., 4 different siRNAs for each gene pooled in 2 siRNA combinations) has been adopted also in a second metastatic FADS1/2 expressing human cell line and the new data have been added in the revised form of the manuscript. These new data are presented in the new **Figures 1-4 and EV1-EV5, Appendix S1-S3**, described and discussed accordingly in the revised form of the manuscript.

It is somewhat frustrating that many of the assays throughout this paper do not include the control cell lines (67NR or BT20) for key assays, where having a non-responsive control would be helpful to show the specificity of the inhibitors for the highly metastatic/aggressive cells.

We apologize for this frustrating aspect. Since the control cells were insensitive to ferroptosis induction we assumed that it was not necessary. However, we understand the request of the Reviewer and we have included the key assays for the requested (67NR, BT20 – included in the initial version of the manuscript) and additional (murine D2A1 and human HCC1937 – included in this round of revision) control cell lines. These data have been shown in **Appendix S1,S2** and commented accordingly in the new version of the manuscript.

The in vivo experiment in Figure 2L should have had analyses of the tumours using a variety of assays such as looking for lipid droplets, lipidomics etc to assess the mechanism by which RSL3 decreased tumour growth. Comparison with 67NR cells in some of the in vitro (Figure 2J/k assays) as well as this in vivo assay with RSL3 would also have been helpful to show that this is specific to the more aggressive lines.

We thank the Reviewer for the interesting observation. Comparison with 67NR in some of the *in vitro* assays has been performed as described in the previous point. We have also performed *ex vivo* analysis of the *in vivo* material to show the effects of RSL3 (such as reactive oxygen species and lipid peroxidation levels) in the 4T1 tumors when compared to the vehicle-treated ones. RSL3-treated 4T1 tumors showed increased levels of malondialdehyde (MDA, one of the final products of polyunsaturated fatty acids peroxidation in the cells) and reduced protein reduction (i.e., accumulation of oxidatively damaged proteins) together with reduced levels of PLIN2 expression, that we have recently shown to be a proxy for lipid droplets content (Bacci, Lorito et al., *Science Trans Med* 2024; PMID: 38416843). The new data have been included in **Figure 2M-O**.

Care should be taken throughout to not overstate the data for 4T07 and 4T1. 4T07 had no significant change in basal respiration on Seahorse or using the Oroboros data, and the text should reflect that these changes were only consistent in the 4T1 cells. It is unclear how Supp Fig S1c-e show "enhanced oxidative metabolism" for 4T07, as their basal OCR does not appear different to 67NR (better quantified in f-h). There is no significant potentiation in 4T07 cells with either inhibitor as suggested: "Although 4T07 and 4T1 cells were already sensitive to RSL3, impairing LD biogenesis using the DGAT1 and DGAT2 inhibitors potentiated the cell death induced by RSL3, particularly in the more aggressive 4T1 cells (Figure 3d)."

We have carefully revised the manuscript, which underwent extensive rephrasing to moderate the assertiveness of the message and to describe the experimental outcomes in a manner that avoids overstating the significance of our findings.

CAY10566 inhibition is missing from Fig S3A and Erastin is should be included in Fig S3B/C analyses.

We thank the Reviewer for the suggestion. These data have been included in the new version of the manuscript in **Figure 4B and EV4A-B**.

The authors should access the lipidomics data published as part of the Xiao et al *Cell Research* 32, 477-490 (2022) manuscript. They should compare their data in Figure 4 with these data to determine whether there is a lipid signature similar to human patient data.

The requested analysis would have held relevance in our study. We retrieved the data from the study conducted by Xiao et al. Unfortunately, the methodology and library utilized for metabolomic profiling in the Xiao study differ substantially from ours, presenting a significant constraint that precludes direct comparison. Nevertheless, we have identified certain lipid species (specifically ceramides) enriched within the 4T1 subtype that exhibit high relevance in the LAR subtype. The inability to directly compare human metabolomic profiles and the noteworthy overlap observed between some lipid species identified in the LAR subtype and those in 4T1 cells could be discussed in the revised manuscript if the Reviewer believes could add value to the study.

The *in silico* analysis should be extended to examine the various TNBC subsets as per the Yang et al 2023 *Cell Metabolism* paper. Are FADS1 and FADS2 higher in the LAR subset? What about the other genes examined? Correlation analyses could also be performed for each of the key genes examined in this study to determine whether they are co-regulated.

This has been a highly constructive comment and has strengthened the message of our study. By retrieving the transcriptomic profile of the Yang et al cohort of TNBC we managed to investigate the contribution (i.e., expression levels, correlation analysis) of FADS1 and FADS2 in the various TNBC subtypes. Relevant Kaplan-Meier analyses of other key players involved in the metabolic profile described (i.e., PLIN2 and ACSL4) have been inserted in the revised manuscript and discussed accordingly (**Figure 2B-D and Figure EV2B**).

Referee #2

(Comments on Novelty/Model System for Author):

This study builds upon previous research, that has demonstrated the heightened sensitivity of triple-negative breast cancers (TNBCs) to ferroptosis. While this aspect has been established for several years, it alone may not provide the desired novelty. The intriguing connection between ferroptosis sensitivity and FADS1/2 deserves attention; however, the data presented in this study is perplexing and primarily relies on siRNA-based approaches, which do not assess the transient effect or its sufficiency in inducing the expected lipidomic changes. Therefore, the inclusion of knockout models is crucial to establish firm conclusions.

We thank the Reviewer for the interesting observation. We have significantly revised the original study, incorporating a substantial amount of new data, notably a model featuring stable downregulation of FADS1 and FADS2. This pertinent model underwent comprehensive lipidomic analysis, which has been included and described in the revised version of the manuscript. Furthermore, we re-established FADS1/2 expression in this knockdown model and conducted functional analyses, conclusively affirming the significance of FADS1 and FADS2 in the process of ferroptosis (refer to subsequent sections for comprehensive elucidation).

Referee #2 (Remarks for Author):

This paper elucidates a mechanism that renders aggressive triple-negative breast cancer cells susceptible to ferroptosis, a form of cell death triggered by lipid peroxidation. The study highlights the significance of FADS1/2 desaturases in regulating intracellular levels of polyunsaturated fatty acids (PUFAs), which are crucial for lipid peroxidation and found in elevated levels in aggressive triple-negative breast cancers. The findings propose that FADS1/2-mediated lipid metabolic reprogramming drives ferroptosis sensitivity in triple-negative breast cancer. While the study offers valuable insights, addressing the following concerns would provide stronger support for the data and claims presented.

Validation of RSL3 as a GPX4 Inhibitor in vivo:

Given the known instability of RSL3 in vivo, the utilization of RSL3 as a GPX4 inhibitor in in vivo experiments needs reconciliation. The authors should provide validation that RSL3 can effectively act as a GPX4 inhibitor in vivo, potentially through liproxstatin rescue as a viable alternative.

Although we did not perform a liproxstatin rescue *in vivo* experiment, we have performed *ex vivo* analyses of the *in vivo* material to show that RSL3 effectively acts as a GPX4 inhibitor in the RSL3-treated 4T1 tumors when compared to the vehicle-treated ones. RSL3-treated 4T1 tumors showed increased levels of malondialdehyde (MDA, one of the final products of polyunsaturated fatty acids peroxidation in the cells) and reduced protein reduction (i.e., accumulation of oxidatively damaged proteins) together with reduced levels of PLIN2 expression, that we have previously shown to be a proxy for lipid droplets content (Bacci,

Lorito et al., *Science Transl Med* 2024; PMID: 38416843). The new data have been included in the revised form of the manuscript (**Figure 2M-O**).

Unquestionable Proof of FADS1/2 Function:

As the central focus of this work, the presented data concerning FADS1/2 requires further substantiation. To establish an incontrovertible role for FADS1/2, the authors should employ FADS1/FADS2 double deficient cells along with corresponding add-back controls. It is crucial to test the titration of RSL3/Erastin over a range of doses rather than a single point to obtain comprehensive evidence. Furthermore, utilizing such models would allow the authors to validate the on-target effects of the inhibitors used in the study. While not mandatory for this study, establishing the role of FADS1/2 in in vivo models would provide additional support.

This has been a constructive comment that further strengthens the message of our study.

We performed a stable downregulation of FADS1 and FADS2 through the simultaneous introduction of 3 shRNAs targeting FADS1 and 3 shRNAs targeting FADS2 using a PiggyBac transposon-mediated shRNA transfer that allowed the generation of stably transfected cells. This model underwent comprehensive lipidomic analysis, which has been included in the revised manuscript and illustrated in **Figure 5A-D** (and described accordingly).

Furthermore, we re-established FADS1/2 expression (using the PiggyBac transposon-mediated transfer) in this knockdown model and conducted functional analyses, conclusively affirming the significance of FADS1 and FADS2 in the process of ferroptosis. Crucially, modulating FADS1 and FADS2 has a direct implication in ferroptosis susceptibility using both erastin and RSL3. These new data have been included in **Figure EV5I-K**.

Partial downregulation of FADS1 and FADS2, utilizing transient and stable approaches, presents both advantages and limitations when compared to total knockout strategies, such as those involving CRISPR technology. While our study exclusively focuses on downregulation rather than total knockout of FADS1 and FADS2, the resulting phenotypic changes and subsequent impact on ferroptosis susceptibility were evident. Partial downregulation may not eliminate the enzyme(s) activity, potentially leading to incomplete phenotypic alterations or confounding results. However, employing both transient and stable transfection methods provided complementary insights into the dynamic regulation of fatty acid desaturation mediated by FADS1/2. Transient transfection allows for rapid assessment of short-term effects, offering immediate response snapshots, whereas stable transfection enables investigation of longer-term consequences and facilitates the establishment of stable cell lines for prolonged studies. Both approaches, in our study, revealed that FADS1/2 are necessary for the ferroptosis execution induced by RSL3 and erastin in the metastatic TNBC model.

Characterization of Lipidome in FADS-Deficient Cells:

To reinforce the proposed role of FADS1/2, it is imperative to characterize the lipidome of FADS-deficient cells in the cell lines where the authors claim its involvement. This analysis would provide valuable insights into the lipid composition changes associated with the alleged role of FADS1/2 in the studied cell lines.

This has been now included in **Figure 5A-D** as claimed in the previous point and described accordingly in the Results section.

Validation of FADS1/2 Function in Human Breast Cancer Cell Lines:

Strengthening the conclusions can be achieved by validating the role of FADS1/2 in a few human breast cancer cell lines. By demonstrating consistency across multiple cell lines, the findings would gain broader applicability and enhance the overall robustness of the study.

We thank the Reviewer for the suggestion. Most of the study has been reproduced in 6 human cell lines (as detailed in Rev 1 point 3). These new data are presented in the new **Figures 1-4 and EV1-EV5, Appendix S1-S3**, described and discussed accordingly in the revised form of the manuscript.

Exploring Potential Interactions with ER Antagonists:

A recent report has proposed that ER antagonists sensitize ER+ breast cancer to ferroptosis by downregulating MBOAT1 (PMID 37267948). Considering the straightforward use of ER antagonists, the authors should explore whether such a mechanism impacts the FADS1/2 axis as well. Investigating this potential interaction would provide additional insights and enrich the study's scope.

We have investigated this aspect in accordance with the Reviewer's suggestion. However, although ER antagonists promote sensitivity to ferroptosis in our ER+ breast cancer cells as well, it appears that no implication of such mechanisms may be pertinent in the ER+ models analyzed employing the standard endocrine targeting methodologies. These additional data have been included in **Appendix S3** and elaborated upon in the main body of the text.

In-depth Discussion on the Advantages of Increased PUFA in TNBC Growth:

While the authors briefly discuss the potential advantages of increased PUFA levels in triple-negative breast cancer growth, a more comprehensive and thoughtful discussion on this aspect is warranted. Such a discussion would promote further discourse on the reasons why cells with heightened sensitivity to ferroptosis thrive, adding depth to the findings and stimulating additional research avenues.

We have discussed the advantages of increased PUFA in TNBC in the Discussion section.

Addressing these concerns would strengthen the authors' data and claims, augmenting our understanding of triple-negative breast cancer and ferroptosis.

Referee #3 (Comments on Novelty/Model System for Author):

In the manuscript, the authors identify FADS1/2 is highly expressed in TNBC. after the related analysis (lipidomic, metabolic, and other assays), The authors found that FADS1/2 high-expressing TNBC are susceptible to ferroptosis. Combination of targeting FADS1/2 and ferroptosis-inducing agents may provide new therapy for TNBC patients. This study provided a new therapy for TNBC patients. However, there are too many issues which need to be addressed.

1. The whole manuscript is difficult to follow. So, the authors should make it smooth and easy to be understood.

We thank the Reviewer for the interesting observation. The manuscript has been extensively rewritten and reorganized and should now be easier to follow.

2. the data quality needs to be improved (e.g., fig.1d). There are too many data which is not so convincing or consistent.

Figure 1D (now **Figure EV1B**) and additional not consistent data have been repeated and consolidated.

3. For the cell models (Fig 1a), the authors should provide the data to support their aggressive ability.

While comprehensive descriptions and characterizations of the aggressive ability of 4T1 and D2A1 models are available, and relevant References have been included in the manuscript, we have additionally acquired data about the 4T1 series (i.e., tumor volume), which we present below. We believe this inclusion may not be essential for the ultimate manuscript; however, it could be added if the Reviewer believes could add value to the study.

4. the authors should provide a figure to summary their finding in Fig.6.

We thank the Reviewer for the suggestion. A figure that summarizes our findings has been drawn and has been inserted as Graphical Abstract of the manuscript (**Figure 7F**).

5. The manuscript should enhance the novelty.

The majority of the experiments outlined in the study have now been replicated across 6 human cell lines (**Figures 1-4 and EV1-EV5, Appendix S1-S3**). The transient downregulation of FADS1/2, conducted in both human and murine metastatic models, and the consequential validation of FADS1/2 relevance in the process of ferroptosis have been substantiated through stable knockdown of FADS1/2 followed by their subsequent reinsertion (**Figures EV5A-K**). Crucially, we additionally introduced FADS1 and FADS2 in

the not-metastatic 67NR cells, thus confirming that FADS1/2 expression is sufficient to confer ferroptosis sensitivity (**Figures EV5L,M**). Moreover, to further explore the biological significance of our findings, although not requested by any of the Referees, we investigated the effects of exogenous fatty acid supplementation in human TNBC cells. We observed that supplementation with PUFA sensitized cells to ferroptosis induction (mediated by both RSL3 and erastin), while MUFA exhibited a protective effect (**Figures 6E,F**). Furthermore, we demonstrated that inhibition of lipid droplet biogenesis enhances ferroptosis-induced cell death in both mouse and human models (**Figures 7D-E**). Finally, the *in vitro* findings have been augmented with *in vivo* data, and the clinical significance has been verified through analysis of pertinent TNBC repositories (**Figure 2**). We believe that the novelty of the manuscript is now reinforced.

Referee #3 (Remarks for Author):

suited to publication as a short report in EMBO Molecular Medicine, but need to be improved.

As a result of the revision process, we have expanded the number of panels and figures. Consequently, we feel that a short report may no longer be suitable for our study.

6th May 2024

Dear Andrea,

Thank you for submitting your revised study. Your manuscript was sent back to the three initial reviewers. As you will see below, they are satisfied with the revisions, and I will therefore be able to accept your manuscript once the following points will be addressed:

1/ Please address the remaining concerns from referees #1 and #3.

2/Manuscript text:

- Please remove the highlights in the text, and only keep in track changes mode any new modification.
- Please remove the one sentence summary.
- Materials and Methods should be after the Discussion. Please address the following:
 - o Mouse models: provide the housing and husbandry conditions for the mice. State details of authority granting ethics approval (IRB or equivalent committee(s)). Include a statement of compliance with ethical regulations.
 - o Statistics: please include a statement on blinding and randomization.
- Data availability: Thank you for providing a reviewer token. Please note that the data must be public before acceptance of the manuscript. Please provide a URL.
- Author contributions: CRediT has replaced the traditional author contributions section because it offers a systematic machine-readable author contributions format that allows for more effective research assessment. Please remove the Authors Contributions from the manuscript and use the free text boxes beneath each contributing author's name in our system to add specific details on the author's contribution. More information is available in our guide to authors.
- "Conflict of interest" should be renamed "Disclosure statement and competing interests": we request authors to consider both actual and perceived competing interests. Please review the policy <https://www.embopress.org/competing-interests> and update your competing interests if necessary.
- Reference format: 10 author names should be listed before et al and DOIs should be removed for published works. References should be before figure legends.

3/ Figures and Appendix:

- Main and EV figures should be uploaded as individual, high resolution figure files.
- Please provide exact p values, not a range, in the figures or in their legends.
- Tables S1 and S2 should be renamed Dataset EV1 and Dataset EV2; both files need their titles and legend added to the file in a separate tab
- The appendix figures should stay compiled in a PDF but their legends should be removed from the manuscript text and added to the file; please correct the nomenclature to "Appendix Figure S1" etc.; the appendix also needs a table of contents with page numbers
- Please note that Dataset EV2 is not referenced in the manuscript text.
- Please address the queries from our data editors in the figure legends:
 1. Please note that the figure legend 7f is missing in the manuscript. This needs to be rectified.
 2. Please indicate the statistical test used for data analysis in the legends of figures 5a, f-g.
 3. Please note that information related to n is missing in the legend of figure 2c.
 4. Although 'n' is provided, please describe the nature of entity for 'n' in the legends of figures 2e, g-i; 3b-e; 4a-c, e-g, 6a-f; 7d-e; EV 1a-e, h; EV 2a, c-g; EV 3a-c; EV 4a-g; EV 5a-d, e-f, h-i, k, m.
 5. Please note that scale bar and its definition are missing for figure 2o.
- As per your email, please indicate in the figure legends and in the material and methods the re-use of loading controls. Please mention the exact samples/proteins.

4/ Thank you for providing Source Data. Please carefully check the labelling of the files (for instance for Fig. 2B, 4E, 4F, 5B, EV2B (empty)).

5/ Checklist:

- please fill in the section 'Experimental animals/'housing and husbandry conditions'.
- please fill in the full section 'Experimental study design and statistics'.
- please fill in the right column for 'sample definition and in-laboratory replication'.
- please fill in the right column for 'Ethics'.
- please fill in the right column for 'Data availability'. Please check that you need to fill in the second line, 'human clinical and genomic datasets'.

6/ Please provide 'The paper explained': EMBO Molecular Medicine articles are accompanied by a summary of the articles to

emphasize the major findings in the paper and their medical implications for the non-specialist reader. Please provide a draft summary of your article highlighting

7/ Synopsis:

The synopsis text should contain both the short stand first and the bullet points: do you confirm the following text?

"The availability of intracellular polyunsaturated fatty acids depends on FADS1/2 desaturases, expressed at higher levels in aggressive triple-negative breast cancers (TNBC) highly susceptible to ferroptosis.

- Fatty acid desaturases 1 and 2 (FADS1/2) are highly expressed in a subset of TNBC with a poorer prognosis.
- FADS1/2 high-expressing TNBC are susceptible to ferroptosis-inducing agents.
- FADS1/2 ablation decreases PUFA/MUFA ration and renders TNBC insensitive to pro-ferroptosis agents.
- Lipid droplets maintain PUFA/MUFA intracellular balance and when targeted enhance cell death induced by ferroptosis induction."

Thank you for providing a nice synopsis image. However, when resized to the required format (550px wide x 300-600px high), the text is not legible. Could you please adjust with bigger font to retain readability?

8/ As part of the EMBO Publications transparent editorial process initiative (see our Editorial at <http://embomolmed.embopress.org/content/2/9/329>), EMBO Molecular Medicine will publish online a Review Process File (RPF) to accompany accepted manuscripts.

This file will be published in conjunction with your paper and will include the anonymous referee reports, your point-by-point response and all pertinent correspondence relating to the manuscript. Let us know whether you agree with the publication of the RPF and as here, if you want to remove or not any figures from it prior to publication.

I look forward to receiving your revised manuscript.

With kind regards,

Lise

***** Reviewer's comments *****

Referee #1 (Comments on Novelty/Model System for Author):

Authors have adequately addressed the previous issues around the models used.

Referee #1 (Remarks for Author):

The authors have comprehensively addressed all my concerns, and this has resulted in a more definitive paper with increased relevance and translatability. There are a couple of minor points that should be considered of a more editorial nature: Radioactive assay - please add the exact time of the overnight incubation to method.

All figure legends should state the numbers of independent experiments performed, there data were often missing.

Quantitation for ACLY and FASN expression (western blot data) across repeat experiments should be shown for Figure 1C

(relative to actin or HSP90). Same applies for other western blots in Figure 1 and 2.

SCD levels are variable in the human cell lines. This should be better detailed in the section around Line 468, especially since SCD1 inhibitors are used in this study. The new text ignores SCD and focusses on FADS1 and FADS2 where the data better align with the mouse models. In addition, there is no mention of SCD expression in the clinical datasets, which should have been examined, or at least commented on if it does not show any significant correlations.

Referee #2 (Remarks for Author):

The authors have largely addressed the comments from the previous revision. I have nothing more to add this stage.

Referee #3 (Comments on Novelty/Model System for Author):

The authors improved the manuscript and answered all my questions. Minor:

1. Fig.3A: GPX4 inhibits PL-PUFA-OOH/ROS? Please correct it.

***** Editor's comments *****

1/ Please address the remaining concerns from referees #1 and #3.

Done. See below

2/Manuscript text:

- Please remove the highlights in the text, and only keep in track changes mode any new modification.
- Please remove the one sentence summary.
- Materials and Methods should be after the Discussion. Please address the following:
 - o Mouse models: provide the housing and husbandry conditions for the mice. State details of authority granting ethics approval (IRB or equivalent committee(s)). Include a statement of compliance with ethical regulations.
 - o Statistics: please include a statement on blinding and randomization.
- Data availability: Thank you for providing a reviewer token. Please note that the data must be public before acceptance of the manuscript. Please provide a URL.
- Author contributions: CRediT has replaced the traditional author contributions section because it offers a systematic machine-readable author contributions format that allows for more effective research assessment. Please remove the Authors Contributions from the manuscript and use the free text boxes beneath each contributing author's name in our system to add specific details on the author's contribution. More information is available in our guide to authors.
- "Conflict of interest" should be renamed "Disclosure statement and competing interests": we request authors to consider both actual and perceived competing interests. Please review the policy <https://www.embopress.org/competing-interests> and update your competing interests if necessary.
- Reference format: 10 author names should be listed before et al and DOIs should be removed for published works. References should be before figure legends.

The above editorial suggestions have been implemented in the revised version of the manuscript.

3/ Figures and Appendix:

- Main and EV figures should be uploaded as individual, high resolution figure files.

Done.

- Please provide exact p values, not a range, in the figures or in their legends.

P-values have been now added in each figure and the statistical comparison are explicit.

- Tables S1 and S2 should be renamed Dataset EV1 and Dataset EV2; both files need their titles and legend added to the file in a separate tab

Done.

- The appendix figures should stay compiled in a PDF but their legends should be removed from the manuscript text and added to the file; please correct the nomenclature to "Appendix Figure S1" etc.; the appendix also needs a table of contents with page numbers

Done.

- Please note that Dataset EV2 is not referenced in the manuscript text.

Done.

- Please address the queries from our data editors in the figure legends:

1. Please note that the figure legend 7f is missing in the manuscript. This needs to be rectified.
 2. Please indicate the statistical test used for data analysis in the legends of figures 5a, f-g.
 3. Please note that information related to n is missing in the legend of figure 2c.
 4. Although 'n' is provided, please describe the nature of entity for 'n' in the legends of figures 2e, g-i; 3b-e; 4a-c, e-g, 6a-f; 7d-e; EV 1a-e, h; EV 2a, c-g; EV 3a-c; EV 4a-g; EV 5a-d, e-f, h-i, k, m.
 5. Please note that scale bar and its definition are missing for figure 2o.
- As per your email, please indicate in the figure legends and in the material and methods the re-use of loading controls. Please mention the exact samples/proteins.

The above data editors' suggestions have been implemented in the revised version of the manuscript.

4/ Thank you for providing Source Data. Please carefully check the labelling of the files (for instance for Fig. 2B, 4E, 4F, 5B, EV2B (empty)).

Done.

5/ Checklist:

- please fill in the section 'Experimental animals/'housing and husbandry conditions'.
- please fill in the full section 'Experimental study design and statistics'.
- please fill in the right column for 'sample definition and in-laboratory replication'.
- please fill in the right column for 'Ethics'.
- please fill in the right column for 'Data availability'. Please check that you need to fill in the second line, 'human clinical and genomic datasets'.

Done.

6/ Please provide 'The paper explained': EMBO Molecular Medicine articles are accompanied by a summary of the articles to emphasize the major findings in the paper and their medical implications for the non-specialist reader. Please provide a draft summary of your article highlighting

Done.

7/ Synopsis:

The synopsis text should contain both the short stand first and the bullet points: do you confirm the following text?

"The availability of intracellular polyunsaturated fatty acids depends on FADS1/2 desaturases, expressed at higher levels in aggressive triple-negative breast cancers (TNBC) highly susceptible to ferroptosis.

- Fatty acid desaturases 1 and 2 (FADS1/2) are highly expressed in a subset of TNBC with a poorer prognosis.
- FADS1/2 high-expressing TNBC are susceptible to ferroptosis-inducing agents.
- FADS1/2 ablation decreases PUFA/MUFA ration and renders TNBC insensitive to pro-ferroptosis agents.
- Lipid droplets maintain PUFA/MUFA intracellular balance and when targeted enhance cell death induced by ferroptosis induction."

The text has been slightly modified and is provided as a new separate file.

Thank you for providing a nice synopsis image. However, when resized to the required format (550px wide x 300-600px high), the text is not legible. Could you please adjust with bigger font to retain readability?

Done.

8/ As part of the EMBO Publications transparent editorial process initiative (see our Editorial at <http://embomolmed.embopress.org/content/2/9/329>), EMBO Molecular Medicine will publish online a Review Process File (RPF) to accompany accepted manuscripts.

This file will be published in conjunction with your paper and will include the anonymous referee reports, your point-by-point response and all pertinent correspondence relating to the manuscript. Let us know whether you agree with the publication of the RPF and as here, if you want to remove or not any figures from it prior to publication.

I agree with the publication of the RPF and authors checklist.

***** Reviewer's comments *****

Referee #1

Authors have adequately addressed the previous issues around the models used.

The authors have comprehensively addressed all my concerns, and this has resulted in a more definitive paper with increased relevance and translatability.

There are a couple of minor points that should be considered of a more editorial nature:

Radioactive assay - please add the exact time of the overnight incubation to method.

Done

All figure legends should state the numbers of independent experiments performed, there data were often missing.

Done

Quantitation for ACLY and FASN expression (western blot data) across repeat experiments should be shown for Figure 1C (relative to actin or HSP90). Same applies for other western blots in Figure 1 and 2.

Done

SCD levels are variable in the human cell lines. This should be better detailed in the section around Line 468, especially since SCD1 inhibitors are used in this study. The new text ignores SCD and focusses on FADS1 and FADS2 where the data better align with the mouse models.

We agree with the reviewer and we have described the heterogenous expression levels of SCD1 and SCD2 (Page 20).

In addition, there is no mention of SCD expression in the clinical datasets, which should have been examined, or at least commented on if it does not show any significant correlations.

In response to the reviewer's comments, we have included an analysis of the clinical datasets in this point-by-point response. The clinical data analysis did not align precisely with the expression patterns of FADS1 and FADS2, suggesting significant differences in the biology of SCD1 and SCD2 in TNBC.

Indeed, by accessing the transcriptomic profile of the FUSCCTNBC cohort (Jiang et al., 2019), we found that *SCD1* is highly expressed in the LAR phenotype (when compared to the other subtypes) whereas *SCD2* is not. Additionally, no positive correlation between *SCD1* and *SCD2* expression levels is identified neither in the whole cohort of TNBC nor in the LAR subtype. However, *SCD1* and *SCD2* higher expression levels identify a subset of TNBC patients with poorer prognosis (see KM plots). Together, the *in vitro* analysis and mechanistic insights from the current study do not support a major role for *SCD1* and *SCD2* in the observed phenotype. We believe that including the *SCD1* and *SCD2* clinical data in the revised manuscript would not add value and would unnecessarily complicate the text. Nonetheless, we defer to the editor's judgment and can include the data as an additional Appendix if deemed necessary.

Referee #2

The authors have largely addressed the comments from the previous revision. I have nothing more to add this stage.

Referee #3

The authors improved the manuscript and answered all my questions. Minor:
1. Fig.3A: GPX4 inhibits PL-PUFA-OOH/ROS? Please correct it.

Done

31st May 2024

Dear Prof. Morandi,

Thank you for sending your revised files. I am pleased to inform you that your manuscript is accepted for publication and is now being sent to our publisher to be included in the next available issue of EMBO Molecular Medicine!

With kind regards,

Lise
